

# Abrupt termination of the Little Ice Age in the Alps in the mid-19th century: lessons from a multi-proxy tree-ring reconstruction of glacier mass balance

Jérôme Lopez-Saez[1], Christophe Corona[1,2], Lenka Slamova[1,3], Matthias Huss[4,5,6], Valérie Daux[7], Kurt Nicolussi[8], Markus Stoffel[1,3,9],

[1]Climate Change Impacts and Risks in the Anthropocene (C-CIA), Institute for Environmental Sciences, University of Geneva, Geneva, Switzerland
[2]Université Clermont-Auvergne, CNRS Geolab (UMR6042), Clermont-Ferrand, France
[3]Department F.-A. Forel for Environmental and Aquatic Sciences, University of Geneva, Geneva, Switzerland
[4]Laboratory of Hydraulics, Hydrology and Glaciology (VAW), ETH Zurich, Zurich, Switzerland
[5]Swiss Federal Institute for Forest, Snow and Landscape (WSL), Birmensdorf, Switzerland
[6]Department of Geosciences, University of Fribourg, Fribourg, Switzerland
[7]Laboratoire des Sciences du Climat et de l'Environnement, LSCE/IPSL, CEA-CNRS-UVSQ, Université Paris-Saclay, Gif-sur-Yvette, France
[8]Institute of Geography, University of Innsbruck, Austria
[9]Department of Earth Sciences, University of Geneva, Geneva, Switzerland

*Correspondence to*: Jérôme Lopez-Saez (jerome.lopez-saez@unige.ch)

**Abstract.** Glacier mass-balance reconstructions provide a means of placing relatively short observational records into a longer-term context. Here, we use multiple proxies from *Pinus cembra* trees from God da Tamangur combining tree-ring anatomy and stable isotope chronologies to reconstruct seasonal glacier mass balance (i.e. winter, summer and annual mass balance) for the nearby Silvrettagletscher over the last two centuries. The combination of tree-ring width, radial cell wall thickness and $\delta^{13}C$ isotope records provide a highly significant reconstruction for summer mass balance, whereas, for winter mass balance, the correlation was less significant but still robust when radial cell lumen was combined with $\delta^{18}O$ and $\delta^{13}C$ records. Combination of the reconstructed winter and summer mass balances allows quantification of the annual mass balance of Silvrettagletscher, for which *in-situ* measurements date back to 1919. Our reconstruction indicates a substantial increase in glacier mass during the first half of the 19th century and an abrupt termination of this phase after the end of the Little Ice Age. Since the 1860s, negative glacier mass balances have been dominant and mass losses accelerate as anthropogenic warming picks up in the Alps. This abrupt termination of the Little Ice Age cannot be found if the mass balance reconstruction is obtained from the gridded temperature and precipitation fields ($1 \times 1$ km) available for Switzerland since 1763.



## 1 Introduction

One of the most iconic and noticeable consequence of anthropogenic climate change at high elevations is the decrease of snow cover and the mass loss of glaciers (Zemp et al., 2019; Beaumet et al., 2021). In the European Alps, glaciers have been retreating since the Little Ice Age (~1850) (Zemp et al., 2006) and future ice volumes are predicted to be largely reduced (Marzeion et al., 2018; Rounce et al., 2023), with ice losses of alpine glaciers reaching up to 90% by 2100 (Zekollari et al., 2019; Vincent et al., 2019). The ongoing reduction of glacier volumes has very direct, negative implications for water

resources, ecosystems and livelihoods (IPCC, 2022; Huss and Hock, 2018; Immerzeel et al., 2020; Cauvy-Fraunié and Dangles, 2019; Bolibar et al., 2020).

To reduce uncertainties in the quantification of future mass losses and their potential consequences, information on past glacier changes is essential as it allows improving simulations of glacier evolution (Brunner et al., 2019). *In-situ* measurements of glacier mass balance constitute a key element in worldwide glacier monitoring. Open-access historical datasets – like those

available from the World Glacier Monitoring Service (WGMS, 2021) – are crucial for an improved understanding of the glacier mass change and the calibration of models used for projections. The (net) mass balance of a glacier surface is defined as the difference between winter accumulation and summer ablation and is generally acknowledged as the prime variable of interest to monitor and project the state of glaciers and their hydrological contribution under global warming scenarios (Hock and Huss, 2021). However, only few glaciers around the world have long-term, direct mass balance observations as these

measurements require considerable manpower, time, and economic resources to be sustained for a meaningful period of time (Kinnard et al., 2022). Despite recent monitoring efforts, the WGMS database – with more than 200 glacier mass balance series worldwide – contains only few records exceeding 20 years.

Various approaches have been used over the last decades to estimate mass balance over multi-decadal timescales, often relying on remotely sensed data. Studies included the use of gravimetry (e.g., Wouters et al., 2019), the interpretation of series of

multiple Digital Elevation Models (e.g., Dussaillant et al., 2019), altimetry (e.g., Gardner et al., 2013) or glacier length change observations (e.g., Hoelzle et al., 2003). Whereas these approaches provided insights into past changes, the temporal resolution of results does not provide information on the inter-annual variability and the drivers of change in glacier mass balance. Mass balance modelling based on meteorological series (Huss et al., 2008; Nemec et al., 2009) offers an alternative method to infer glacier mass balance over long time-scales at high temporal resolution but results are not backed-up with *in-situ* observations

before the onset of glaciological measurements and therefore might be biased or may incompletely resolve the relevant processes.

Tree-ring proxies clearly have the potential to overcome these limitations and to extend glacier mass balance series farther back in time. Based on the concept of Oerlemans and Reichert (2000) according to which mass balance series can be reconstructed from long meteorological records, several dendrochronological studies have been developed to demonstrate the

reliability of high-elevation tree-ring proxies as reliable recorders of past summer temperature and – to a lesser extent – also winter precipitation (e.g., Büntgen et al., 2005; Coulthard et al., 2021; Carrer et al., 2023; Lopez-Saez et al., 2023). In the



2000s, seminal papers used tree-ring width (TRW) series to reconstruct mass balance patterns for glaciers in Canada (Lewis and Smith, 2004; Larocque and Smith, 2005; Watson et al., 2006). Since 2007, multiproxy mass balance reconstructions combining TRW with maximum latewood density (MXD), stable isotopes or blue Intensity (BI) have been developed for
glaciers in Pacific North America (Wood et al., 2011; Malcomb and Wiles, 2013), Scandinavia (Linderholm et al., 2007; Hiemstra et al., 2022) or Central Asia (Zhang et al., 2019) (see Table 1 for a complete review).

| Reference | Location | Lat | Long | Mass Balance Reconstruction | | | Period | Nb of trees |
|---|---|---|---|---|---|---|---|---|
| | | | | Winter | Summer | Annual | | |
| Nicolussi and Patzelt, 1996 | Gepatschfemer, Tyrol, Austria | 46°84 N | 10°75 E | EW (TRW) | LW (TRW) | EW + LW (TRW) | 1400-1987 | / |
| Lewis and Smith 2004 | Vancouver Island, British Columbia, Canada | 49°40 N | 125°40 W | | | TRW | 1412-1998 | 53 |
| Larocque and Smith 2005 | Mt Waddington, British Columbia, Canada | 62°01 N | 144°32 W | TRW | TRW | TRW | 1550-2000 | / |
| Watson et al., 2006 | Peyto Glacier, Canada | | | TRW | TRW | TRW | 1673-1994 | 74 |
| Linderholm et al 2007 | Storglaciären, Sweden | 67°55 N | 18°35 E | Circulation indices | TRW + MXD | Circulation indices+TRW + MXD | 1780-1981 | / |
| Wood et al 2011 | Place Glacizer, British Columbia, Canada | 50°25 N | 122°36 W | MXD + TRW | TRW | | 1585-2006 | 61 |
| Malcomb and Wiles 2013 | Various glaciers, USA and Canada | 47° N | 123° W | | | TRW +LW + MXD | 1500-1999 | / |
| Shekhar et al 2017 | Various glaciers Western Himalayan, India | 32° N | 77° E | | | TRW | 1615-2015 | 189 |
| Zhang et al, 2019 | Tuyuksuyskiy Glacier, Kazakhstan | 43°03 N | 77°05 E | TRW | Stable isotope | TRW + stable isotope | 1850-2014 | 50 |
| Cerrato et al., 2020 | Careser Glacier, Italy | 46°25 N | 10°41 E | Precipitation records | MXD | Precipitation records + MXD | 1811-2013 | 24 |
| Hiemstra et al., 2022 | Jotunheimen, Norway | 61°6 N | 8°3 E | Precipitation records | BI | Precipitation records + BI | 1722-2017 | 32 |

Table 1. Synthesis of existing dendroclimatic studies and tree-ring proxies as well as meteorological data used for the reconstruction of the winter, summer and annual glacier mass balances.

In the Alps, mass balance reconstructions are much scarcer: Nicolussi and Patzelt (1996) reconstructed 600 years of glacier mass balance for the Gepatschferner glacier using TRW records. More recently, Cerrato et al., (2020) reconstructed fluctuations of the Careser glacier (Italian Alps) since 1811 using MXD series and HISTALP meteorological records (Auer et al., 2007) to reconstruct summer and winter mass balance, respectively.

In Switzerland, GLAMOS (GLAcier Monitoring Switzerland; www.glamos.ch) hosts a complete compilation of measured and
re-analyzed mass balance data of Swiss glaciers, of which several span much of the 20[th] century (Huss et al., 2015). Despite the uniqueness of these records and their potential for validation and calibration of proxy-based reconstructions, no attempts have been undertaken so far to extend these datasets beyond the 20[th] century or even to preindustrial times.

In addition, recent developments in quantitative wood anatomy (QWA), relying on the analysis of dimensions of wood cells in tree-rings, demonstrated that this approach offers an unparalleled measurement precision and substantial gain in temperature
reconstructions (Lopez-Saez et al., 2023; Seftigen et al., 2022; Allen et al., 2022; Björklund et al., 2023). It is thus the aim of this study to reconstruct the seasonal mass balance (i.e. winter and summer mass balance) of Silvrettagletscher, Eastern Swiss Alps, employing stable isotope ($\delta^{18}O$, $\delta^{13}C$) and tree-ring anatomy chronologies of *P. cembra* which has recently been shown to be very sensitive to mean temperature over the ablation season (April–September; Lopez-Saez et al., 2023). Using increment cores from trees growing close to Silvrettagletscher, we (i) construct TRW, anatomical and isotope chronologies to (ii) derive
annual time series of past summer temperature and winter precipitation as proxies of summer and winter glacier mass balance. This provides new insights into mass balance dynamics of Alpine glaciers during the maximum and termination of the Little Ice Age, a phase of important dynamics in glacier evolution but very limited direct evidence on the rates and the exact timing of changes.



## 2 Materials and methods

**2.1 Glacier and tree-ring sites**

The study focuses on the Eastern Swiss Alps, close to the borders with Austria and Italy (Fig. 1A). Silvrettagletscher (46.85° N; 10.08°E) is a small temperate mountain glacier with a surface area of presently 2.7 km$^2$ extending from 3,070 down to 2,470 m asl (Figure 1A, B). The mean equilibrium line altitude of the glacier is at 2,820 m asl and its first mass balance measurements date back to 1919 CE. Seasonal observations at two stakes were conducted until 1959, when the stake network was increased
to 40 stakes. Today, 18 stakes are surveyed seasonally. Huss et al. (2009) re-analyzed and homogenized the seasonal stake data back to 1959.

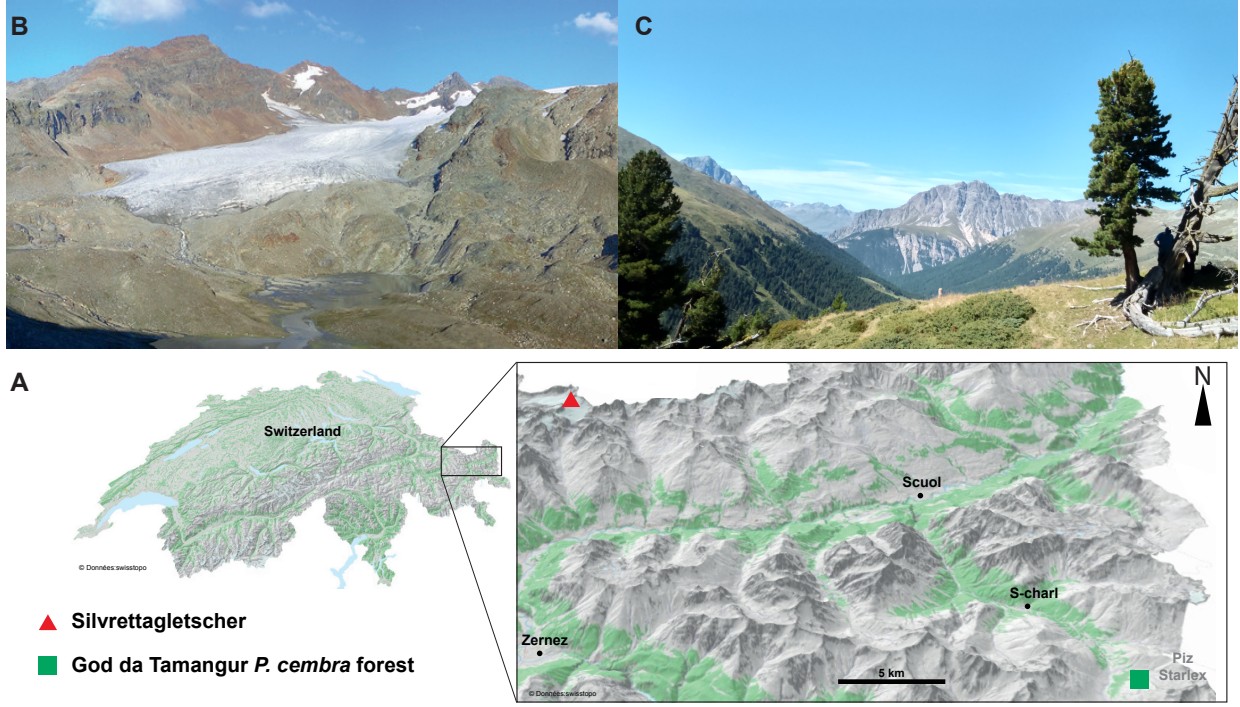

Figure 1. (A) The study site is located in the eastern part of Switzerland, close to the municipality of Scuol. (B) Overview of Silvrettagletscher (www.glamos.ch) and (C) detailed view of a century-old *P. cembra* tree from God da Tamangur (Val S-
charl, Scuol, Grisons, Switzerland) selected for analysis.



Silvrettagletscher is a global reference glacier of the WGMS and the monitoring is maintained in the frame of GLAMOS. The tree-ring site is located c. 30 km to the southwest of Silvrettagletscher and known locally as God da Tamangur (46.68°N; 10.36°E) – meaning the « forest back there» in Vallader Romansch. It is the highest, pure, and continuous *P. cembra* forest in
Europe (Figure 1A,C). The forest is located at an elevation of up to 2,300 m asl, at the end of Val S-charl (Grisons, Switzerland), on the NW-facing slope of Piz Starlex (3,075 m asl). Lopez-Saez et al. (2023) recently showed that various wood anatomical parameters extracted from this forest allow robust reconstruction of past temperature variability at annual to multidecadal timescales.

## 2.2 Sample collection and wood processing

Tree cores were collected during a field campaign in summer 2018. To perform wood cell anatomical measurements, 20 trees were sampled using a 12 mm increment borer. From each tree, we extracted two increment cores at breast height (*c.* 130 cm above ground). Ring widths were measured to the nearest 0.01 mm using TSAPWin (Rinntech, Germany), cross-dated using standard dendrochronological procedures (Stokes and Smiley, 1996) and checked for dating and measurement errors with the COFECHA software (Holmes, 1983). Ring widths from single radii were summarized to mean widths per tree before values
from individual trees were averaged into a master TRW chronology.

## 2.3 TRW and wood anatomical analyses

In a subsequent step, the 12 mm cores were split into 4–5 cm long pieces to obtain 15 μm thick cross-sections with a rotary microtome (Leica RM 2255/2245). The sections were stained with Safranin and Astra blue to increase contrast and fixed with
Canada balsam following standard protocols (Gärtner and Schweingruber, 2013; von Arx et al., 2016). Digital images of the microsections - at a resolution of 2.27 pixels/μm - were produced at the Swiss Federal Research Institute WSL (Birmensdorf, Switzerland), using a Zeiss AxioScan Z1 (Carl Zeiss AG, Germany). For the 20 selected trees, we used the ROXAS (v3.1) image analysis software (von Arx and Carrer, 2014) to automatically detect anatomical structures for all tracheid cells and annual ring boundaries for the period 1800–2017. We excluded measurements of samples with cell walls damaged during
sampling or preparation and focused on two parameters in quantitative wood anatomy analyses: radial cell lumen diameter ($D_{rad}$) and radial cell wall thickness ($CWT_{rad}$) (Prendin et al., 2017; von Arx and Carrer, 2014).

Following Lopez-Saez et al. (2023), we assigned each cell to tangential bands of 40 μm in radial width (with distances measured parallel to ring boundaries). In addition, we determined the transition from earlywood to latewood cells for each ring using Mork's index = 1, at a 10 μm radial resolution (Denne, 1989; see Lopez-Saez et al., 2023 for more details). For each
ring, maximum values of $D_{rad}$ and $CWT_{rad}$ were extracted from the bands identified as belonging to the ring. For $D_{rad}$, maximum values were extracted from each ring in the earlywood. For $CWT_{rad}$, maximum values were extracted from the latewood.





The conventional TRW measurements were detrended with a negative exponential function to eliminate non-climatic (e.g., age-related growth trends and other biological disturbances) effects from the series (Fritts, 1976; Cook and Kairiukstis, 1990). The detrended series were then aggregated into a TRW chronology using a biweight robust mean which reduces the influence

of outliers (Cook and Peters, 1981). Given the absence of any evident long-term ontogenetic trend in anatomical series, detrending is not normally considered necessary (Carrer et al., 2018; Lopez-Saez et al., 2023) in QWA studies. In a next step, empirical measures of dendroclimatic signals (Hughes et al., 2011) were computed to test the strength of the environmental information embedded in the chronologies using the maximum overlap of pairwise correlations (Bunn et al., 2013). These included average inter-series correlation (RBAR$_{EFF}$) and expressed population signal (EPS) (Wigley et al., 1984). All analyses

were performed in R Studio (R Studio Team, 2020) using the R package dplR (Bunn, 2008; Bunn et al., 2013).

## 2.4 Isotopic analyses

For the isotopic analyses ($\delta^{18}$O and $\delta^{13}$C), we selected 10 trees that were between 242 and 634 years old at the time of sampling. Cores were cut ring by ring with a scalpel at the Laboratoire des Sciences du Climat et de l'Environnement (LSCE, Gif-sur-Yvette, France). The wood from each ring was processed separately between 1968 and 2017 and every fifth year between 1802 and 1967, so that in total, 73 years were measured on each individual core. For all other years between 1802 and 1965, material

from the 10 trees of the same year was pooled prior to analysis. The wood samples were grounded and α-cellulose was extracted according to the SOXHLET chemical method (Leavitt and Danzer, 1993) and homogenized ultrasonically. The oxygen and carbon isotopic composition was obtained by high temperature pyrolysis in a high-temperature conversion elemental analyzer (Thermo Scientific) coupled to an Isoprime mass spectrometer (see Penchenat et al., 2022 for details). The measured sample

values were corrected based on an internal laboratory reference of cellulose (Whatmann® CC31) analyzed every three samples in each sequence of analysis. The analytical precisions of the instruments were within ±0.20‰ for $\delta^{18}$O and ±0.10‰ for $\delta^{13}$C, respectively, based on the standard uncertainty of the mean. A correction to the $\delta^{13}$C raw series was applied by means of linear interpolation to compensate for decreasing $\delta^{13}$C in organic matter related to fossil-fuel combustion and increasing atmospheric concentration (Francey et al., 1999; McCarroll and Loader, 2004). The oxygen and carbon isotopic composition were expressed

as δ following:

$$\delta = (R_{samp} - R_{VSMOW}\text{-}1) \times 1000$$

where $R_{Samp}$ is the isotopic ratio in the sample and $R_{VSMOW}$ the isotopic ratio in the Vienna Standard Mean Ocean Water (for

oxygen) or the Vienna Pee-Dee belemnite (for carbon) (Coplen, 1996).



## 2.5 Meteorological series

In this study, the gridded (1 × 1 km) daily mean temperature and precipitation time series available from Imfeld et al. (2023), hereafter referred to as *Imfeld23*, were used to both identify the main drivers of radial growth and to reconstruct glacier mass

balance fluctuations. The dataset (1763-2020) includes meteorological data rescued by various initiatives (Brugnara et al., 2020; Pfister et al., 2019; Brugnara et al., 2022) for the late 18[th] and early 19[th] centuries and systematic measurements available in Switzerland since 1864. Time series were initially checked and homogenized on a subdaily basis (following Brugnara et al., 2020). The dataset was then reconstructed at a 1 × 1 km resolution using an analogue method, which resamples meteorological fields for a historical period based on the most similar day in a reference period. The fields were improved with data

assimilation for temperature and bias correction with quantile mapping for precipitation (Imfeld et al., 2023). Several limitations must be considered when working with this exceptional dataset: (1) the reconstruction skills decrease prior to 1864 as fewer stations provide direct observations, (2) larger reconstruction errors are observed for precipitation than for temperature due to the heterogeneous nature of precipitation, and (3) the quality of the dataset is spatially heterogeneous and considerably reduced in the Alps and the southern side of the Alps due to both the scarcity of observations and more complex topo-climatic

conditions.

## 2.6 Climate–growth relationships

In a first step, we correlated wood proxy data with *Imfeld23* by selecting the grid point centered over the God da Tamangur study site. To test for the robustness of the mixed proxies series for climate–growth relationship, we calibrated regression models on temperature and precipitation averaged over 30 to 330-day windows starting on October 1 of the year preceding

ring formation (n-1) and ending on September 30 of the year in which the ring was formed (n) using the R package DendroTools (Jevšenak and Levanič, 2018). This time window was chosen according to the growing season of *P. cembra* trees in the Alps (Saulnier et al., 2011). Correlations were computed over the 1802-2017 period covered by the tree-ring proxy series.

## 2.7 Glacier mass balance multi-proxy reconstructions

In a next step, wood proxies correlated with fall/winter precipitation and spring/summer temperatures were used as predictors

to reconstruct winter ($B_w$) and summer ($B_s$) mass balance of Silvrettagletscher, respectively. Glacier-wide time series of winter and summer mass balance available for Silvrettagletscher (1919-2022) from the WGMS (Huss et al., 2015; WGMS, 2021) were used as predictands. Seasonal point measurements acquired in early May and September, respectively, were extrapolated to unmeasured areas of the glacier using a model-based extrapolation technique (Huss et al., 2015). Changes in the glacier geometry due to advance or retreat have a direct impact on the overall mass balance, mainly due to changes in the elevation

range. These effects are included in the observational dataset of glacier-wide mass balances as the latter always refer to the





instantaneous glacier geometry. For the period before 1920, however, we do not explicitly adapt glacier geometry but assume the relations derived and tested over a 100-year period with significant changes to be representative.

A principal component regression (PCR) approach was chosen to reconstruct winter and summer mass balances. The first $n$ principal components (PCs) with eigenvalues >1 were retained as predictors to develop a multiple linear regression model. We
computed 10,000 summer and winter mass balance reconstructions using a split calibration–verification procedure coupled with a bootstrap approach in which 50% of the years covered by both the mass balance observation and wood-proxy datasets were randomly extracted for calibration while the remaining years were used for validation over the 1920-2017 period. For each sampling, the root mean-square error (RMSE), coefficient of determination ($r^2$ for the calibration and $R^2$ for the verification periods), reduction of error (RE) and coefficient of efficiency (CE) statistics (Cook et al., 1995) were applied to
test the predictive capacity of the transfer function. Calibration and validation statistics are illustrated for each of the TRW, wood anatomical and isotope parameters with their 5th, 50th and 95th percentiles. In a final step, annual mass balance ($B_a$) was reconstructed using the sum of the final winter and summer mass balance reconstructions.

In parallel, we reconstructed winter and summer mass balances over the 1763-2020 period using the gridded temperature and precipitation field records from *Imfeld23*. With the purpose to identify the optimal time window for the reconstruction, we
selected the grid-point closest to Silvrettagletscher and calibrated regression models between daily temperature series computed over one-to-365-days windows starting on January 1 of the year preceding observations of summer mass balance and ending on December 31 of the year in which the mass balance measurement was acquired. In addition, daily precipitation-temperature series were used as regressors for observed winter-summer mass balances. For each optimal precipitation and temperature time window identified, we computed 10,000 reconstructions following the calibration/verification procedure
described for tree rings.

## 3 Results and discussion

### 3.1 Isotope and wood anatomical features chronology characteristics

We measured wood anatomical features for the period 1802–2017 CE on all samples for a total of 75–100 radial files per ring,
with anatomical information catalogued by its position in each dated tree ring. After the exclusion of cells with walls damaged during sampling or preparation, a total of 2,277,779 tracheid cells were used for analysis.

Figure 2 showcases the evolution of wood anatomical parameters as a function of relative distance to ring border. It also shows that, based on Mork's index, latewood represents roughly 10% of total ring width on average. *P. cembra* trees feature the classical ring structure of conifers growing in cold, temperate environments, with an increase in radial cell wall thickness
($CWT_{rad}$, Figure 2A), and a monotonic reduction in radial diameter ($D_{rad}$, Figure 2B), from earlywood to latewood. Statistical characteristics of the Tamangur chronologies are summarized in Table 2.



The EPS and Rbar values show that TRW has a stronger common signal (Rbar=0.39, EPS=0.85) than the wood anatomical chronologies ($D_{rad}$, $CWT_{rad}$), with the Rbar for the latter ranging between 0.16 (for $D_{rad}$ at 40 μm radial band width) and 0.25 (for a $CWT_{rad}$ at 40 μm radial band width). Regarding isotope parameters, Rbar values computed for 10-yr windows are 0.46 ($\delta^{13}C$) and 0.48 ($\delta^{18}O$). Several studies report lower common signals in wood anatomical parameters than in TRW series from deciduous (Fonti and García-González, 2004) and conifer trees (Seftigen et al., 2022).

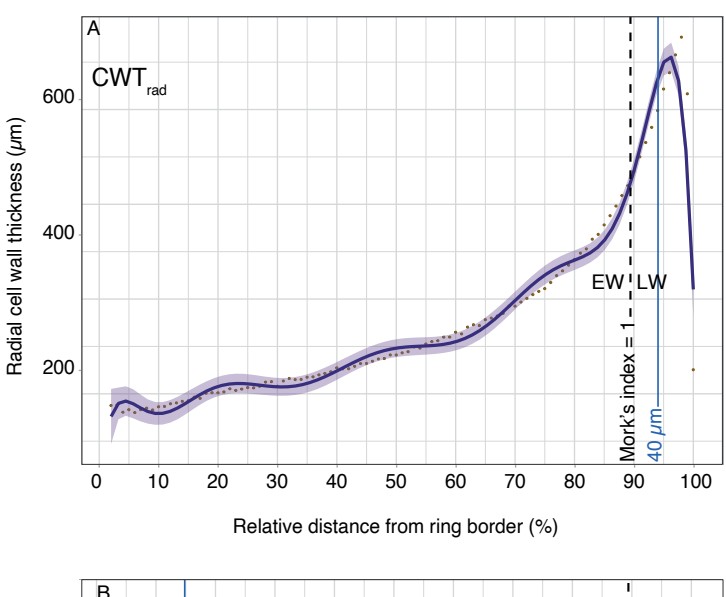

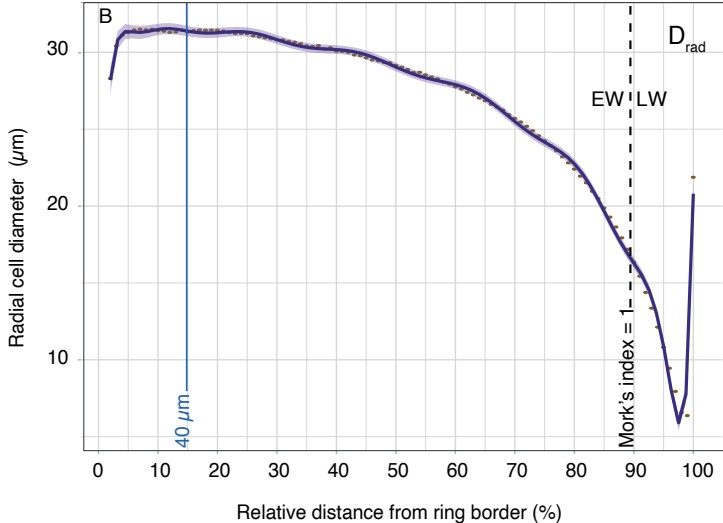

Figure 2. Profiles of (A) radial cell-wall thickness ($CWT_{rad}$) and (B) radial cell diameters ($D_{rad}$) along *P. cembra* tree rings. Black lines represent the mean values of twenty trees over 217 years (1800–2017), whereas the shadowed purple areas delimit the 95 % confidence interval. The blue line represents maximum values for each of the wood parameters analyzed for 40 μm wide radial bands. The dotted black line shows the mean relative position of the transition between earlywood and latewood according to Morck's index = 1.





This weaker signal is generally attributed to inter-annual variability in microscopic wood features (Olano et al., 2012; Liang et al., 2013; Pritzkow et al., 2014), heterogeneous intra-annual internal physiological processes which regulate carbon assimilation and allocation in tree rings (Eilmann et al., 2006; Fonti and García-González, 2004; Balanzategui et al., 2021) or to relationships with intra-annual environmental variables (Yasue et al., 2000; Ziaco et al., 2016) rather than to limiting factors exerted over the entire growing season (Eckstein, 2004; Ziaco et al., 2016).

| Wood proxy | Bandwith (µm) | EPS | Rbar | r | Temperature | Precipitation |
|---|---|---|---|---|---|---|
| Tree-ring width (TRW) | | 0.85 | 0.39 | 0.4* | 05/14 (n) - 08/01 (n) | |
| | | | | -0.31* | | 05/04 (n) - 07/27 (n) |
| Radial diameter ($D_{rad}$) - Earlywood | 40 | 0.75 | 0.16 | 0.46* | 03/08 (n) - 09/10 (n) | |
| | | | | 0.23* | | 11/05 (n-1) - 03/5 (n) |
| Radial cell wall thickness ($CWT_{rad}$) - Latewood | 40 | 0.84 | 0.25 | 0.68* | 07/15 (n) - 09/11 (n) | |
| | | | | -0.28* | | 06/12 (n) - 10/10 (n) |
| $\delta^{13}C$ (10-yr interval) | | 0.9 | 0.46 | -0.43* | 06/12 (n-1) - 05/06 (n) | |
| | | | | -0.22* | | 05/26 (n) - 07/25 (n) |
| $\delta^{18}O$ (10-yr interval) | | 0.91 | 0.48 | 0.44* | 04/11 (n) - 09/13 (n) | |
| | | | | -0.25* | | 11/30 (n-1) - 08/08 (n) |

Table 2. Statistics of chronologies based on TRW, $D_{rad}$, $CWT_{rad}$ and $\delta^{13}C$, $\delta^{18}O$ as well as optimal time windows used in annual mass balance reconstructions. EPS = expressed population signal, Rbar = average inter-series correlation, r = coefficient of correlation and their significance levels (p) at * p≤0.05. For details see text.

## .2 Climate-growth relationships

Correlation coefficients between the TRW chronology from Tamangur and the gridded temperature and precipitation fields
from *Imfeld23* show that spring-to-summer temperature is the main driver of radial growth ($r_{max}$= 0.4, p<0.05; May 14-August 1) between 1802 and 2017 CE (Table 2, Figure S1A). For wood anatomical parameters, significant correlations exist between radial diameter ($D_{rad}$) and early spring to late summer temperatures (r=0.46, p<0.05; March 8 – September 10). This association spans a longer seasonal-window than the ones reported by Carrer et al. (2017, 2018) for earlywood cell areas of *P. cembra, P. abies* and *L. decidua* trees in the Italian Alps which are restricted to mid-May/early June, mid-June/mid-July and mid-May/late-
August, respectively. For this parameter, significant, albeit lower, correlations (r=0.23, p<0.05,) are also found with November (n-1) to early March (n) precipitation. Studies focusing on xylem features in alpine conifers (Carrer et al., 2017, 2018) did neither include the year preceding ring formation nor show comparable correlation profiles with fall/winter precipitation. Yet, in the White Mountains of Arizona (U.S.), Balanzategui et al. (2021) showed that lumen diameter of *P. menziesii* growing at high elevation sites (2,500-2,900 m asl) indeed positively encode signals from past September – current February precipitation.
In our case, highest correlations were obtained between the $CWT_{rad}$ chronology and temperatures over a 152-day time window extending from July 15 to September 11 (r=0.68; p<0.05, Figure 3A). R values computed between $CWT_{rad}$ and summer temperatures agree with results reported by Carrer et al. (2018) for the Italian Dolomites (r>0.6 with July 15–August 15 temperature; 1926–2014) or Știrbu et al. (2022) for the Carpathians (r=0.65 with July-August temperatures; 1961–2013). The



period overlaps with the wall-thickening phase observed in latewood during summer for high elevation *P. abies* trees (Gindl et al., 2001; Rossi et al., 2008). Its duration also agrees with current knowledge on xylogenesis, which can last from 1 (mild environments) to 2 months (cold environments) in latewood cells (Rossi et al., 2008; Cuny et al., 2013; Castagneri et al., 2017). Mean daily temperature from June (n-1) to May (n) (r=-0.43, p<0.05) and during the growing season (April-September, r=0.44, p<0.05) are the main drivers of $\delta^{13}$C and $\delta^{18}$O variations. A negative correlation is also found between $\delta^{13}$C and May 26–July 25 (n) (r=-0.22) and between $\delta^{18}$O and fall (n-1) to summer (n) precipitation totals (r=-0.25) (Table 2). Both chronologies also portray a significant association with winter precipitation, positive for $\delta^{13}$C and negative for $\delta^{18}$O (Figure S1B). Analysis of $\delta^{13}$C and $\delta^{18}$O stable isotope signals in *P. cembra* trees has been initiated only recently in the Alps (Haupt et al., 2014; Arosio et al., 2020) and in the Carpathians (Nagavciuc et al., 2021, 2022; Kern et al., 2023). In the Alps, studies have focused on the detection of age-related trends in the series (Arosio et al., 2020) but did not provide correlation profiles with climatic variables. In the Carpathians, by contrast, positive correlations were reported with April-August temperatures $\delta^{18}$O (Nagavciuc et al., 2021), in line with our results. By contrast, no significant correlation was found between $\delta^{13}$C and temperature. Consistent results are also found with precipitation for $\delta^{13}$C, both in Tamangur and Carpathian chronologies which show a negative correlation with June precipitation. The winter signal embedded in the $\delta^{18}$O and $\delta^{13}$C chronologies of God da Tamangur echoes the significant associations observed between isotope series and winter precipitation in the Arctic (Holzkämper et al., 2008), Kazakhstan (Qin et al., 2022) , the Tibetan Plateau, northwestern China (Grießinger et al., 2017; Wernicke et al., 2017; Liu et al., 2013; Qin et al., 2015), Iran (Foroozan et al., 2020) or Pakistan (Treydte et al., 2006) and are thus interpreted as the result of the trees' use of precipitation from before the growing seasons stored in the soil or in groundwater reservoirs.

### 3.3 Multi-proxy glacier mass balance reconstruction

Based on this preliminary climate–growth relation analysis, we tested several combinations of parameters to reconstruct winter ($B_w$) and summer ($B_s$) glacier mass balances (Table 3).

| Multi-proxy reconstructions | r2 | RE | CE | Wood proxies |
|---|---|---|---|---|
| Summer Mass Balance (SMB) | [0.35-0.58] 0.47* | [0.21-0.56] 0.43 | [0.15-0.55] 0.4 | TRW - Cwtrad |
| Winter Mass Balance (WMB) | [0.1-0.31] 0.2* | [-0.18-0.22] 0.1 | [-0.29-0.2] 0.05 | δ18O - δ13C - Drad |

Table 3. Statistics of Summer Mass Balance (SMB) and Winter Mass Balance (WMB) reconstructions based on combination with TRW, CWT$_{rad}$ (for SMB) and $\delta^{18}$O, $\delta^{13}$C, D$_{rad}$ (for WMB) and their significance levels (p) at * p≤0.05.

For $B_w$, the best combination of proxies sensitive to winter precipitation includes D$_{rad}$, $\delta^{18}$O and $\delta^{13}$C. The first two PCs of the multiple regression are positively correlated with winter mass balance fluctuations ($r^2$=0.2, p<0.05; Table 3) and allow a statistically significant reconstruction (RE=0.1, CE=0.05) over the period 1920-2017 (Figure 3A) covered by glaciological measurements and tree-proxy reconstructions. In detail, the 30-yr moving correlations computed between the reconstructed and observed winter mass balance show an abrupt decrease of r values for time windows ending between 1975 and 2000





(r<0.25, Figure 4A). Interestingly, during most of this time period (1984-2003) no *in situ* measurements of winter mass balance
were available (Huss and Bauder, 2009) and the gaps in the winter mass balance series were filled using a calibrated mass
balance model driven by data from nearby meteorological stations (Huss et al., 2015).

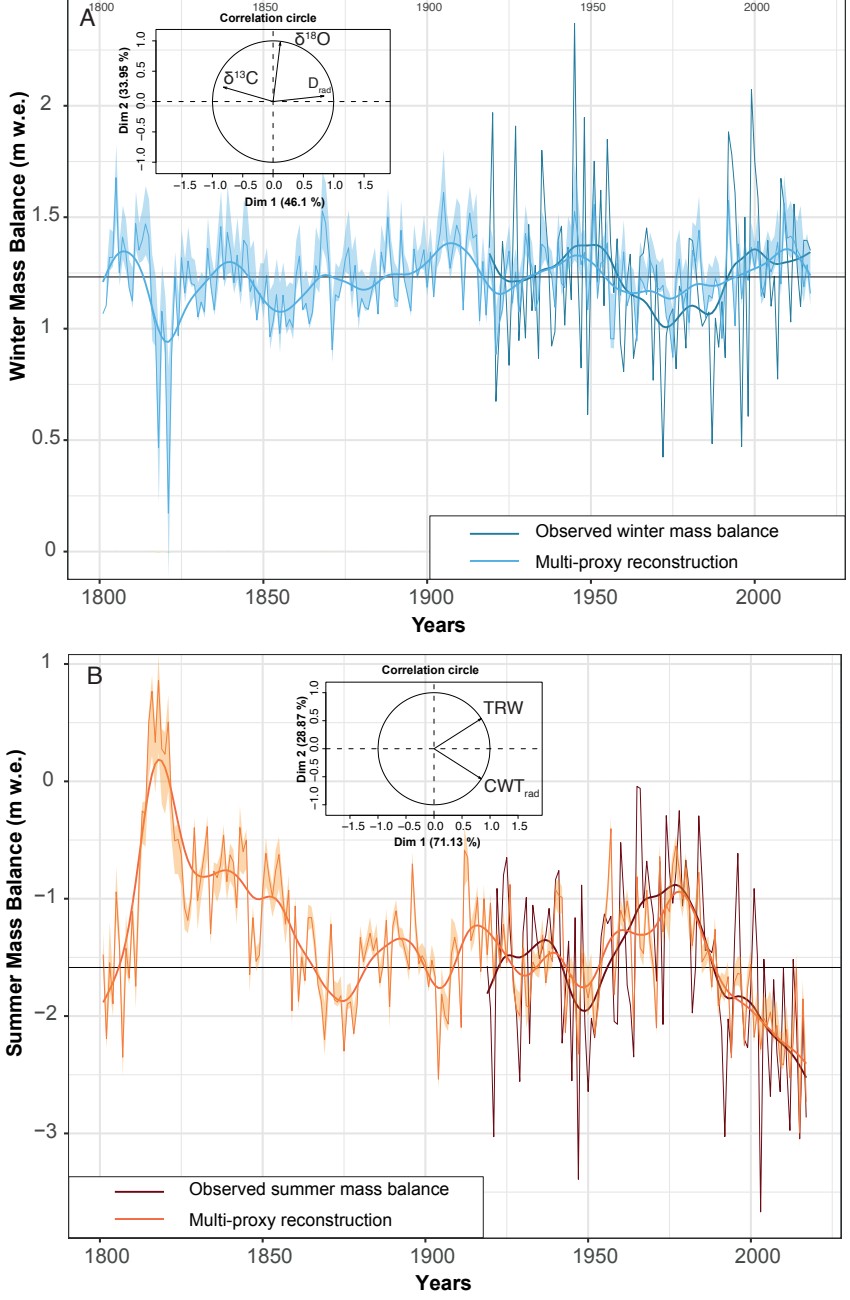

Figure 3. (A) Comparison of interannual and decadal (i.e. smoothed with a 11-yr spline) observed (A) winter mass balance
and (B) summer mass balance as obtained with the multiple wood-proxy reconstruction over the period 1802–2017.


We can therefore hypothesize that the decrease in correlation could be linked to the quality of the mass balance time series and not necessarily to the tree-proxy dataset. At decadal timescales, the 11 yr-spline smoothed winter mass balance reconstructions correlate at 0.65 with observations and capture positive (i.e. from the late 1940s to the late 1950s) and negative (i.e. from the 1960s to 1980s) anomalies over the entire period (Figure 3A).

The best combination of summer temperature-sensitive proxies for summer glacier mass balance include the TRW and $CWT_{rad}$ chronologies (Figure 3B). The two first principal components (PCs) of the PCR allow a robust reconstruction of $B_s$ ($r^2=0.47$, $p<0.05$) and significant RE (0.43 [0.21-0.56]) and CE (0.4, 0.15-0.55]) statistics (Table 3). These values exceed those computed by Cerrato et al. (2020) using *P. cembra* MXD series for Ghiacciaio del Careser over the period 1967-2005 ($r^2=0.45$). The summer balance reconstruction also depicts the positive (1960s to early 1980s) and negative (i.e. 1950s and since the late

1980s) anomalies found in the measurements (Figure 3B). By contrast to Cerrato et al. (2020) who report a reduction of $B_s$ prediction skill for *P. cembra* based on MXD records in the early 1980s, the 30-year correlations obtained from the observed and multi-proxy reconstructed summer mass balance time series remain >0.48 throughout the period and show limited standard deviation (0.06) between 1920 and 2017 (Figure 4A).

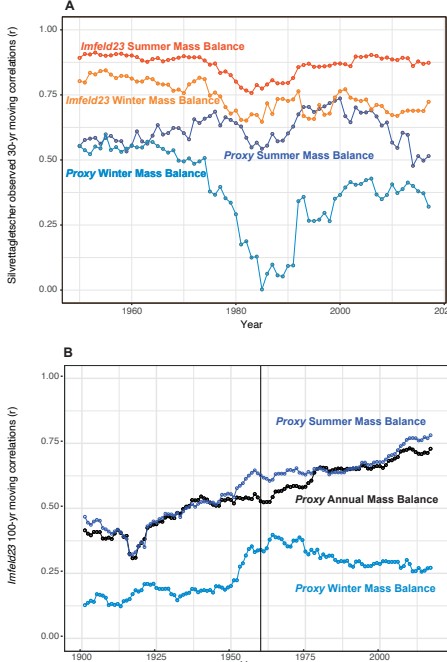


Figure 4. (A) 30-yr moving correlations (r) of summer and winter mass balances estimated from gridded temperature and precipitation fields (Imfeld et al., 2023) and multiple wood-proxy summer and winter mass balance reconstruction (this study) with the Silvrettagletscher winter and summer mass balance records (Huss et al., 2015). (B) 100-yr moving correlations of proxy summer, winter and annual mass balance with *Imfeld23* (Imfeld et al., 2023).





Figure 5. Comparison between annual (thin grey lines) and 11-yr spline smoothed (thick lines) variations of (A) the multiple wood-proxy annual mass balance reconstruction, (B) observed annual mass balance and (C) annual mass balance reconstructed from gridded temperature and temperature fields (Imfeld et al., 2023). Periods with positive mass balance are shown in blue, periods with negative mass balance are given in red.

These results confirm that $CWT_{rad}$ records do neither suffer from divergence (Lopez-Saez et al., 2023) nor from standardization issues which notoriously affect both TRW and MXD records (Cook et al., 1995; Björklund et al., 2019).

Considering the annual glacier mass balance reconstruction, we finally combined proxy values for winter and summer balance (Figure 5A), thus extending the GLAMOS record by 120 years. At interannual timescales, Pearson correlation coefficients between the wood-proxy annual glacier mass balance and *in-situ* measurements at Silvrettagletscher are highly significant at r=0.62 (p<0.05). When applying a 11-yr spline to both the reconstructed and measured time series, correlations reach r=0.87. The influence of proxy winter balance on proxy annual balance is limited, explaining only 8% of the annual average variance.



These results are consistent with Zemp et al. (2015) for glaciers in the European Alps where winter mass balance explains 6% of the annual mass balance variations on average.

### 3.4 Comparison with *Imfeld23* records and reconstructions

For the mass balance of Silvrettagletscher, correlations computed between the observed glacier mass balance and *Imfeld23* are higher than those computed with the wood proxies. The optimal time window for reconstruction extends from May 17 to September 19 for summer mass balance ($r^2$=0.77, p<0.05; 1920–2017) and from September 22 (n–1) to May 2 (n) for winter mass balance ($r^2$=0.50, p<0.05). At the annual scale, the *Imfeld23* reconstruction explains 74% of annual glacier mass balance variability observed over the period 1920-2017. Over the 1802-2017 period, the summer, winter and annual wood-proxy reconstructions significantly correlate (r=0.63, 0.15 and 0.61,p<0.05) with the reconstruction based on *Imfeld23* (Figure 5C). Synchronous periods are characterized by positive anomalies in the 1810s, 1840s, 1910s and the late 1970s and likewise, negative anomalies are observed in the two timeseries in the 1870s, the early 20th century, the 1950s and since the mid-1980s. This is well in line with contemporary and documentary sources as well as information on dated moraines available for the Swiss Alps (e.g., Zumbühl et al., 2008; Nussbaumer and Zumbühl, 2012; Schimmelpfennig et al., 2014). More interestingly, the proxy reconstruction shows a strong glacier mass increase in the first part of the 19th century and confirms the abrupt mass loss previously reported in the Alps in the 1850s and 1860s considered to correspond to the end of the Little Ice Age (Holzhauser et al., 2005; Vincent, 2005; Zemp et al., 2006; Painter et al., 2013). Sigl et al. (2015, 2018) hypothesized that the mass gain in the Alps in the early 19th century could result from the strong negative radiative forcing induced by at least five large tropical eruptions between 1809 and 1835 (Sigl et al., 2015; Toohey and Sigl, 2017) in tandem with the Dalton solar minimum (Usoskin et al., 2013; Jungclaus et al., 2017). The positive mass balance of Silvrettagletscher between 1810 and 1850 confirms, for the first time and at annual resolution, the consequences of the volcanic forcing and the solar minimum on Alpine glaciers.

By contrast to the wood-proxy reconstruction, this very positive mass balance pattern is not reproduced by the *Imfeld23* reconstruction and, therefore, the moving correlations computed for 100-yr time periods between both records thus decrease significantly before the 1860s (Figure 4B). This divergence between the two reconstructions during preindustrial times is thought to result from the limited robustness of the *Imfeld23* dataset prior to 1864, and the complete absence of high-elevation records available at that time. It is therefore likely that the gridded temperature and precipitation fields fail to reproduce changes in winter precipitation distributions in the early stages of the reconstruction. Yet, winter precipitation is considered to represent yet another potential driver of glacier mass balance fluctuations until at least 1875 (Vincent et al., 2005). Furthermore, the limited agreement between the reconstructions during the maximum of the Little Ice Age indicate that factors other than temperature and precipitation, as provided by the *Imfeld23* dataset, are relevant to correctly describe variations in glacier mass balance. The reduction in radiative forcing is likely key for mass balance and its effect might be better resolved by wood-proxy reconstructions than by meteorological variables alone.




**Conclusions**

Mountain glaciers are reliable and unequivocal indicators of climate change due to their sensitivity to changes in temperature and precipitation (Zhang et al., 2019). The advance or retreat of a glacier is thereby related to the amount of snow accumulation as well as snow and ice melt, commonly referred to as its mass balance. This study allowed development of multi-proxy

chronologies from *P. cembra* wood traits based on the dynamic relationships between climate processes that jointly influence tree (cell) growth and glacier mass balance. The Silvrettagletscher has been monitored since 1919 and therefore forms a very robust baseline against which the wood-proxy reconstruction presented here can be compared. The study also constitutes an important first step in extending glacier mass-balance records beyond the instrumental period for Switzerland and throughout the European Alps. Results of this work likewise provide novel insights into the dynamics of glacier mass gain and loss during

the final stages and the maximum of the Little Ice Age in the early- to mid-19th century. By combining wood anatomical parameters and stable isotopes, we obtain very promising results for seasonal glacier mass balance reconstructions, because some proxies are sensitive to mean temperatures over the entire ablation period while others estimate winter precipitation during the accumulation period. Our results based on multiple wood-proxies reveal that glacier mass gains during the final stages of the Little Ice Age were strongest between 1810 and 1820, most likely as a result of reduced radiative forcing and not

just as a result of unusual temperature and/or precipitation. This period of positive mass balances and resulting glacier advances rapidly ended in the 1860s and 1870s when a first episode of substantial negative mass balances led to a first phase of "modern" glacier downwasting.

**Competing interests**

The contact author has declared that none of the authors has any competing interests.

**Acknowledgements**

J.L.-S., C.C., and M.S. acknowledge support from the Swiss National Science Foundation (SNSF) Spark project "MNEMOSYNE" and a scnat research grant from the Research Commission of the Swiss National Park (FOK- SNP). C.C.,

J.L.-S., L.S. and M.S. received funding from the Swiss National Science Foundation (SNSF) Sinergia project CALDERA (no. CRSIIS_183571).

**Figure captions**


Figure 1. (A) The study site is located in the eastern part of Switzerland, close to the municipality of Scuol. (B) Overview of Silvrettagletscher (www.glamos.ch) and (C) detailed view of a century-old *P. cembra* tree from God da Tamangur (Val S-charl, Scuol, Grisons, Switzerland) selected for analysis.





Figure 2. Profiles of (A) radial cell-wall thickness (CWT$_{rad}$) and (B) radial cell diameters (D$_{rad}$) along *P. cembra* tree rings. Black lines represent the mean values of twenty trees over 217 years (1800–2017), whereas the shadowed purple areas delimit the 95 % confidence interval. The blue line represents maximum values for each of the wood parameters analyzed for 40 μm wide radial bands. The dotted black line shows the mean relative position of the transition between earlywood and latewood according to Morck's index = 1.


Figure 3. (A) Comparison of interannual and decadal (i.e. smoothed with a 11-yr spline) observed (A) winter mass balance and (B) summer mass balance as obtained with the multiple wood-proxy reconstruction over the period 1802–2017.

Figure 4. (A) 30-yr moving correlations (r) of summer and winter mass balances estimated from gridded temperature and 410 precipitation fields (Imfeld et al., 2023) and multiple wood-proxy summer and winter mass balance reconstruction (this study) with the Silvrettagletscher winter and summer mass balance records (Huss et al., 2015). (B) 100-yr moving correlations of proxy summer, winter and annual mass balance with *Imfeld23* (Imfeld et al., 2023).

Figure 5. Comparison between annual (then grey lines) and 11-yr spline smoothed (thick lines) variations of (A) the multiple 415 wood-proxy annual mass balance reconstruction, (B) observed annual mass balance and (C) annual mass balance reconstructed from gridded temperature and temperature fields (Imfeld et al., 2023). Periods with positive mass balance are shown in blue, periods with negative mass balance are given in red.

Table 1. Synthesis of existing dendroclimatic studies and tree-ring proxies as well as meteorological data used for the 420 reconstruction of the winter, summer and annual glacier mass balances.

Table 2. Statistics of chronologies based on TRW, D$_{rad}$, CWT$_{rad}$ and δ$^{13}$C, δ$^{18}$O as well as optimal time windows used in annual mass balance reconstructions. EPS = expressed population signal, Rbar = average inter-series correlation, r = coefficient of correlation. For details see text.


Table 3. Statistics of Summer Mass Balance (SMB) and Winter Mass Balance (WMB) reconstructions based on combination with TRW, CWT$_{rad}$ (for SMB) and δ$^{18}$O, δ$^{13}$C, D$_{rad}$ (for WMB).

**Supplementary Material**

Figure S1. (A) Correlations of the raw radial cell-wall thickness (CWT$_{rad}$) and spline detrended ring width (TRW) chronologies with air temperatures reconstructed with the Imfeld et al. (2023) dataset over time windows ranging from 121-to 273-day



windows. (B) Correlations of the raw $\delta^{18}O$, $\delta^{13}C$ and $D_{rad}$ chronologies and *Imfeld23* precipitation sums over time windows ranging from 274 (n-1) to 120-day windows.

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
