# Peer review of "Multiproxy tree-ring reconstruction of glacier mass balance: Insights from *Pinus cembra* trees growing near Silvretta glacier (Swiss Alps)"

_EGUsphere, 2023_

## Referee Comment (RC1)

**GENERAL COMMENT**

In the manuscript 'Abrupt termination of the Little Ice Age in the Alps in the mid-19[th] century: lessons from a multi-proxy tree-ring reconstruction of glacier mass balance' Lopez-Saez and co-authors present seasonal (and annual) mass balance reconstructions for a Swiss glacier since 1802 CE. Authors use several proxies obtained by different methods (total ring width, quantitative wood analysis, and isotopes) and Principal Component Analysis to perform a multiparameter linear regression. The obtained loadings were used to explain and to reconstruct mass balances' variance in the last century (since 1919). Results are statistically significant and pass the tests normally used in dendroclimatological reconstructions. They show variations of the mass balance compatible with known glaciological history in the Alps. Thus, authors conclude that the use of different wood-proxies permits the seasonal mass balance reconstruction of the Silvretta glacier.

The manuscript, in my opinion, is well written and the aims are clearly presented. Authors present exceptional datasets for an overlooked species in the Alps (i.e., *Pinus cembra*). In fact, in my knowledge, they present first isotope chronologies from Swiss stone pine in the area and one of the firsts chronologies of anatomical traits. Scientific design is solid and well presented. Moreover, only few dendroglaciological papers about European Alps were published, thus the manuscript is also characterized by a high level of novelty. However, in my opinion, the use of some methodologies is partly questionable, and both discussion and conclusion lack a bit of control in some parts resulting presented in a bloated fashion and quite speculative way.

**SPECIFIC COMMENTS**

[#001] Line 25: $\delta^{13}C$ isotope is not utilized for $B_s$ reconstruction.

[#002] Lines 35–36: Consider mentioning that glaciers are one of the Essential Climate Variables (ECV) as reported by the Global Climate Observing System (GCOS) (https://gcos.wmo.int/en/home)

[#003] Lines 57–66: Authors at lines 57–61 state that '*Mass balance modelling based on meteorological series (Huss et al., 2008; Nemec et al., 2009) offers an alternative method to infer glacier mass balance over long time-scales at high temporal resolution but results are not backed-up with in-situ observations before the onset of glaciological measurements and therefore might be biased or may incompletely resolve the relevant processes.*'. At lines 62–66 they state: '*Tree-ring proxies clearly have the potential to overcome these limitations and to extend glacier mass balance series farther back in time. Based on the concept of Oerlemans and Reichert (2000) according to which mass balance series can be reconstructed from long meteorological records, several dendrochronological studies have been developed to demonstrate the reliability of high-elevation tree-ring proxies as reliable recorders of past summer temperature and – to a lesser extent – also winter precipitation (e.g., Büntgen et al., 2005; Coulthard et al., 2021; Carrer et al., 2023; Lopez-Saez et al., 2023).*'. These sentences seem in conflict to me. In the former authors declare that since there are not glaciological measures, reconstruction based on glaciological modelling and meteorological series might be biased, on the other hand, they declare, in the latter sentence, that tree-ring proxies can overcome to this limitation since they are representative of the summer temperature and winter precipitation that can be used to reconstruct mass balance. Thus, tree-ring proxies were used to reconstruct temperature and precipitation (i.e., meteorological series) that were used to reconstruct the mass balance that, however, supply results that might be biased since glaciological measurements are still missing before the start of the monitoring. Please consider clarifying the concepts.

[#004] Lines 77–78: In Cerrato et al. 2020 the authors did not used HISTALP dataset, but an improved version of the database presented in Brunetti et al. 2006 as described in material and methods.

Brunetti, M., Maugeri, M., Monti, F., and Nanni, T.: Temperature and precipitation variability in Italy in the last two centuries from homogenised instrumental time series, Int. J. Climatol., 26, 345–381, https://doi.org/10.1002/joc.1251, 2006.

[#005] Lines 91–93: In the manuscript only one glacier was tested and no comparison with others mass balance was performed. Thus, referring to 'Alpine glaciers' seems a bit pretentious if it is meant as glaciers of the entire European Alps.

[#006] Line 98: Please consider adding the reference period for which the mean equilibrium line altitude is calculated. If it refers to the entire period (1919–2023) please consider adding also the maximum and the minimum since in the last century alpine glaciers withdraw quite continuously and abundantly.

[#007] Line 108: the tree stand is located c. 30 km southeast of the glacier (not southwest).

[#008] Line 149–150: Please check the total number of analysed rings. Between 1968 and 2017 there are 49 years, whereas between 1802 and 1967 there are 165 years, that divided by 5 results in 33 years (34 if 1802 is considered too). Thus, if I correctly understood, the total number of analysed rings is 83 and not 73 as reported.

[#009] Line 164: $R_{samp}$, samp with lowercase 's' to be coherent with the formula.

[#010] Lines 229–230: Please consider moving the sentence about the isotope at the end of the paragraph (at Line 243) to not interrupt the discussion about the QWA.

[#011] Line 248: the number of the chapter is missing in section heading (i.e., 3).

[#012] Lines 260–261: probable typo, from July 15 to September 11 there are 58 days, not 152 as declared. Moreover, Figure 3A shows the mass balance reconstruction, I think that the figure was moved to supplementary material or totally removed.

[#013] Line 267: see comment [#034].

[#014] Lines 269–270: The sentence is misleading. From Figure S1B very low values of correlation is interpretable, spanning between −0.1 and 0.1. In some situation, as function of the windows width, slightly positive or negative results can be appreciated (usually between (−)0.1 and (−)0.3).

[#015] Line 277: The winter signal embedded in $\delta^{13}C$ chronology is not reported in Table 2 and only partially inferable from supplementary material.

[#016] Line 283–284: In table 3 only one combination for reconstructing $B_s$ and $B_w$ is reported. Consider rephrasing.

[#017] Table 3: I wonder if all isotopes are really necessary in the Principal Component Regression model for $B_w$. More precisely, one isotope chronology (Carbon) resulted to be most sensible to the summer precipitation (i.e., ablation period, Table 2), and the other (Oxygen) is sensible to the precipitation from November to August (covering not only accumulation period but also quite the first two-third of the ablation season). How different would be the results if only $D_{rad}$ is used? Or, in another way, are the authors sure to include proxies that are sensible to environmental parameters of ablation season in the reconstruction of $B_w$? It seems quite counterintuitive to me, and clarification are necessary in my opinion.

[#018] Lines 289–293: Even if it is true that the first two Principal Components (PCs) positively correlate with $B_w$, seems that authors overlooked at the relationships between the original data and the PCs variables. If I correctly understood figure 3, it reported the correlation circle between the original variables (i.e., isotopes and $D_{rad}$) and the first two dimensions obtained by PCA. Looking at that plot seems that only $D_{rad}$ is positively correlated with the PC's first dimension and basically uncorrelated with the second. I can thus hypothesize that, due to the positive correlation index between the PC variables and the $B_w$ (shown in Table 3), $D_{rad}$ is representative of the environmental condition that drive the $B_w$. Contrarily, Carbon isotope show a high negative correlation with PC first dimension (indicating a quite linear negative correlation with this dimension) with a (maybe significant) positive correlation with the second PC dimension. This seems to be coherent with previous analysis, i.e., Carbon isotope find its best correlation window with summer precipitation, and being summer precipitation mostly liquid, they enhance snow melting with heat transfer along the snowpack. Thus, probably, the negative correlation that was showed along the first PC dimension is representative not of a $B_w$, but of a $B_s$ that, judging from figure 3, seems quite well negatively correlated with $B_w$. Considering the Oxygen isotopes, the series result completely uncorrelated with the PC first dimension being aligned with the axis of the second dimension. Considering that first and second dimension have quite the same explanatory power of the original dataset (i.e., 46.1 and 34.0, respectively) it is plausible that Oxygen represents something different from both $D_{rad}$ (precipitation from November to May and spring-summer temperature), and Carbon (summer

precipitation and whole year temperature, this last point is questionable as reported in comments [#015] and [#017]). In fact, from previous analysis it results that Oxygen series is correlated with spring-summer temperature (as well as $D_{rad}$) but with winter-to-summer precipitation. Maybe the representativeness of this variable of such a long precipitation period appoints it as a second major source of variability in the original dataset. Authors should consider these results, or at least supply more explanation on the motivation that drove them to use proxies sensible to summer precipitation and/or to two-third of the ablation season to reconstruct the $B_w$ bearing in mind that correlation does not mean causation.

[#019] Lines 301–302: Authors hypothesis could be true, however should be noted that in the referred period occurred the last phase of positive $B_a$ in the (Southern) Alps (Huss et al. 2015). Moreover, if the gap in $B_w$ starts in 1984, I wonder how it is possible that the correlation values start to decrease 10 years before (as author stated, and as figure 4 shows, considering that the results are right-aligned, thus the considered 30-year window in 1974 is 1945–1974) and reach their lowest value nearby the years when the modelled $B_w$ starts. Moreover, the lowering in correlation values ended around 2000 (i.e., the 30-year right-aligned window 1971–2000) when around 57% of the data are modelled. Considering this, the hypothesis supplied by authors seems to be not really supported by reported data. Probably, the changes in environmental conditions that bring less negative or even positive $B_a$, is not well represented by the selected variables for $B_w$ (it is just a hypothesis that should be verified). On the other hand, a decrease of the correlation values in those years is observable also considering the seasonal mass balance reconstruction based on *Imfeld23* (both $B_s$ and $B_w$) and the wood-proxy based $B_s$. Also in these cases, the lowering in correlation values starts well before the 1984, thus, in my opinion the lack of measured glaciological data could not be the (only) explanation to the observed behaviour in correlation trend.

Huss, M., Dhulst, L., and Bauder, A.: New long-term mass-balance series for the Swiss Alps, J. Glaciol., 61, 551–562, https://doi.org/10.3189/2015jog15j015, 2015.

[#020] Lines 305–307: To me it is not clear the advantage in using all the variables deriving from a PCA instead of the original data. The PCA was thought to lowering the number of considered variables, creating new variables that 'summarize' the variance of the original data. Variables reduction is obtained retaining only those new variables that explain the largest part of the original data variance (usually 80% but it depends by the aims). If all PC variables are kept, it is equivalent to apply the multiparametric regression using the original data.

[#021] Line 308: consider modifying 'Ghiacciaio del Careser' to Careser glacier, as used before in the manuscript.

[#022] Lines 310–313: From figure 4 it is clear that the increasing trend appreciable from 1950 to 2000 (as exception of the 1980s where a decrease in correlation values is appreciable as commented in [#019]), is reverted to a negative trend since 2000 with correlation coefficient that drops from 0.75 to 0.5 in 17 years (mean decrease of -0.015 year$^{-1}$, analysis should be performed to verify if the trends are significant and if the change is significant too, but I can speculate that, at least the negative one, is significant). Considering this, the decreasing of correlation starts in the 1970s, 10 years before the start of decreasing reported in Cerrato et al. 2020 (in Figure 4 the correlation are right-aligned, thus the 2000 value refers to 1971–2000 time window and this is why the decrease in correlation values starts in 1970s).

[#023] Lines 326–327: Being the correlation obtained using a multiple regression, this statement is not supported by data in this context, even if it is true as reported in cited papers. Moreover, consider to cite also Cerrato et al. 2019, that report data about the divergence between Swiss stone pine MXD and temperature in the high-frequency domain and being the source of data for Cerrato et al. 2020.

Cerrato, R., Salvatore, M. C., Gunnarson, B. E., Linderholm, H. W., Carturan, L., Brunetti, M., De Blasi, F., and Baroni, C.: A Pinus cembra L. tree-ring record for late spring to late summer temperature in the Rhaetian Alps, Italy, Dendrochronologia, 53, 22–31, https://doi.org/10.1016/j.dendro.2018.10.010, 2019.

[#024] Lines 351–353: A reconstruction of an Alpine glacier mass balance at annual scale was already reported by Cerrato et al. 2020 and by Nicolussi and Patzelt, 1996 (in my knowledge, but could be other studies. These studies are already cited in the manuscript even if the latter is missing in the reference list) and

both show less negative (or even positive) mass balances around the last peak of the LIA. Please consider rephrasing. Moreover, in the present study, volcanic forcing or radiative data were not considered, thus the sentence, in the present form, seems a bit speculative to me.

[#025] Lines 354–364: Speculative paragraphs. If two reconstructions are available and no verification is possible, it is basically impossible to determinate which is the most correct. It is certainly true that back in time, meteorological series loss representativeness and explanatory power in remote areas, but, considering reported data, also tree-ring proxies reconstruction suffer of a decrease of explanatory power in cold phases (see for instance 1980s where mass balance data, even if modelized, are present) and also after 2000s (even if in this last case the environmental conditions that drive a loss of correlation are hotter than the previously experienced; see comments [#019] and [#022] for more details). Thus, concluding that the $B_a$ based on *Imfeld23* lacks representativeness whereas tree-ring bases reconstruction surely represent the behaviour of the glacier in such bloated form seems a bit speculative. Moreover, authors never consider that their approach using the meteorological data could suffer of a big and simply issue: authors calibrated the reconstruction using an optimal time window based on temperature and precipitation occurred since 1919. Statistics are solid and tests were passed in the considered period. By counterpart, authors are assuming, based on their results, that the length of the accumulation and ablation seasons are the same in a period where the temperature has been proved been lower and also precipitation might be, testified by a different duration of the snowpack (Carrer et al., 2023, cited in the manuscript). Maybe authors should consider that is not the meteorological dataset, but the selected optimal window of a fixed length based on recent environmental conditions that can bias the results, as already reported in Cerrato et al. 2020. However, should be noted that the here proposed reconstruction for $B_w$ shows lower values during the Dalton minimum compatible with previous work that reported more dryer winter during that period (Anet et al., 2014). However, the *Imfeld23* based $B_w$ is not reported, thus it is impossible to evaluate if also meteorological-based $B_w$ reconstruction shows comparable results.

Anet, J. G., Muthers, S., Rozanov, E. V., Raible, C. C., Stenke, A., Shapiro, A. I., Brönnimann, S., Arfeuille, F., Brugnara, Y., Beer, J., Steinhilber, F., Schmutz, W., and Peter, T.: Impact of solar versus volcanic activity variations on tropospheric temperatures and precipitation during the Dalton Minimum, Clim. Past, 10, 921–938, https://doi.org/10.5194/cp-10-921-2014, 2014.

[#026] Lines 372–374: The study can be considered a first step in Switzerland, but not throughout the Alps since both Cerrato et al. 2020 and Nicolussi and Patzelt, 1996 already presented mass balance reconstructions. Consider rephrasing.

[#027] Lines 374–375: Due to the lacks of validation on the correctness of the $B_a$ reconstructions (at the actual state it is impossible to define which is the most correct reconstruction between the wood-proxy based and the *Imfeld23*-based $B_a$ since no comparison with previous reconstruction is performed, neither a comparison between the potential glacier volume with geomorphological and/or cartographical evidence) the sentence seems quite speculative.

[#028] Lines 378–380: This sentence seems speculative. Since it is impossible to validate both wood-based and *Imfeld23* reconstructions (see comments [#025] and [#027]) it is also impossible to be sure of the correctness of the estimated quantity of water equivalent gain (or loss) in period were the reconstructions differ in a more pronounced way (from Figure 4 and 5, the period of less agreement between wood-proxies based and *Imfeld23*-based reconstructions occurred for the entire XIX Century). Moreover, Authors stated earlier that in the earlier portion the used meteorological dataset is not completely reliable, so it is impossible to verify this sentence, maybe the disagreement between the expected and obtained *Imfeld23* $B_a$ is due to the uncertainties of the original dataset, or maybe not.

[#029] Figure 1: Consider inverting the vertical order of the inset A and both B and C.

[#030] Figure 2: Consider to explain the meaning of the purple dotted line (or purple dots, but it seems a line to me) in caption.

[#031] Figure 3: caption, unclear to me, please consider rephrasing.

[#032] Figure 5: please consider to maintain the same y-axis scale among the plots for readability.

[#033] Figure S1: caption is misleading on the time-window information. Moreover, reported information (e.g., standardization method and windows length) does not match those declared in the main manuscript and thus the results are not easily comparable with those reported in the main text. In fine, the addition of contour lines at the significance level of $p<0.05$ will be appreciated.

[#034] Table 2: correlation between $\delta^{13}C$ and temperature: the reported optimal time window is equal in length to the maximum window tested (330 days), probably enlarging the tested windows, other (and longer) 'optimum windows' could be found. Beside this mine speculative consideration, authors in M&M stated that they '*calibrated regression models on temperature and precipitation averaged over 30 to 330-day windows starting on October 1 of the year preceding ring formation (n-1) and ending on September 30 of the year in which the ring was formed (n)*'. Results does not match with declared methods.
Caption: 'optimal time windows used in annual mass balance reconstructions' is misleading. If I correctly understood, these are the optimal time windows resulting from the correlation analysis between the tree-ring parameters and the meteorological series. Mass balances are not involved in these results.

[#035] Table 3: consider using $B_s$, $B_w$, $CWT_{rad}$, $D_{rad}$, $\delta^{13}C$, and $\delta^{18}O$ both in table and in caption to be coherent with the rest of the manuscript.

---

## Author Comment (AC1)

A

Switzerland

▲ Silvrettagletscher

■ God da Tamangur *P. cembra* forest

Scuol

S-charl

Zernez

Piz Starlex

5 km

N

B

C

A. CWT_Rad

Radial cell wall thickness (μm)

Mork's index = 1

40 μm

Earlywood — Latewood

Relative distance from ring border (%)

B. D_rad

Radial cell diameter (μm)

40 μm

Mork's index = 1

Earlywood — Latewood

Relative distance from ring border (%)

**A. Temperature**

**B. Precipitation**

**C. Winter mass balance**

**D. Summer mass balance**

**A.** Winter mass balance

**B.** Summer mass balance

Instrumental series
Multi-proxy reconstruction

A. Multiproxy reconstruction

B. Observations

C. *Imfeld23* reconstruction

| Wood proxy | Bandwith (μm) | EPS | Rbar |
|---|---|---|---|
| Tree-ring width (TRW) | | 0.85 | 0.39 |
| Radial diameter (D$_{rad}$) - Earlywood | 40 | 0.75 | 0.16 |
| Radial cell wall thickness (CWT$_{rad}$) - Latewood | 40 | 0.84 | 0.25 |
| δ13C (10-yr interval) | | 0.9 | 0.46 |
| δ18O (10-yr interval) | | 0.91 | 0.48 |

| Glacier Mass Balance | Wood proxies | RMSE | r2 | RE | CE |
|---|---|---|---|---|---|
| Winter Mass Balance (Bw) | Drad | [343.22-353.31] 345.38 | [0.03-0.2] 0.1 | [-0.12-0.16] 0.08 | [-0.23-0.14] 0.04 |
| | δ18O | [350.47-359.87] 352.44 | [0.01-0.14] 0.06 | [-0.10-0.10] 0.04 | [-0.21-0.08] -0.001 |
| | **δ18O - Drad** | [335.45-347.15] **338.47** | [0.07-0.25] **0.15** | [-0.11-0.20] **0.08** | [-0.22-0.18] **0.06** |
| Summer Mass Balance (Bs) | TRW | [701.07-721.41] 705.59 | [0.05-0.26] 0.14 | [-0.1-0.23] 0.12 | [-0.19-0.20] 0.07 |
| | Drad | [685.76-706.51] 690.2 | [0.07-0.29] 0.17 | [-0.08-0.27] 0.16 | [-0.16-0.25] 0.12 |
| | CWTrad | [585.93-602.33] 589.35 | [0.28-0.51] 0.38 | [0.18-0.51] 0.37 | [0.12-0.5] 0.35 |
| | δ18O | [728.86-755.31] 734.63 | [0.01-0.18] 0.07 | [-0.19-0.12] 0.04 | [-0.29-0.1] -0.002 |
| | TRW - Drad | [669.76-699.32] 677.34 | [0.12-0.34] 0.16 | [-0.09-0.3] 0.16 | [-0.18-0.27] 0.12 |
| | TRW - CWTrad | [555.33-576.75] 560.95 | [0.35-0.58] 0.47 | [0.21-0.56] 0.43 | [0.15-0.55] 0.4 |
| | TRW - δ18O | [689.24-719.44] 697.43 | [0.08-0.31] 0.18 | [-0.16-0.25] 0.11 | [-0.25-0.23] 0.07 |
| | Drad - CWTrad | [584.24-606.58] 589.88 | [0.29-0.53] 0.41 | [0.14-0.52] 0.35 | [0.08-0.5] 0.33 |
| | Drad - δ18O | [657.71-687.24] 665.3 | [0.15-0.38] 0.25 | [-0.07-0.33] 0.19 | [-0.15-0.31] 0.15 |
| | CWTrad - δ18O | [584.65-609.98] 591.26 | [0.30-0.52] 0.41 | [0.11-0.5] 0.34 | [0.08-0.48] 0.32 |
| | **CWTrad - TRW - Drad** | [471.23-601.00] **537.12** | [0.37-0.6] **0.49** | [0.19-0.57] **0.43** | [0.14-0.55] **0.4** |

---

## Author Response (AR1)

**Response to Reviewer #1. Riccardo Cerrato**

Dear Editor,

Below we provide a point-by-point response to the comments of Reviewer 1 that were very helpful to finalize the manuscript. Our responses to the reviewer comments are given here, and new text in the manuscript is pasted in quotation marks.

**GENERAL COMMENT**

In the manuscript 'Abrupt termination of the Little Ice Age in the Alps in the mid-19th century: lessons from a multi-proxy tree-ring reconstruction of glacier mass balance' Lopez-Saez and co-authors present seasonal (and annual) mass balance reconstructions for a Swiss glacier since 1802 CE. Authors use several proxies obtained by different methods (total ring width, quantitative wood analysis, and isotopes) and Principal Component Analysis to perform a multiparameter linear regression. The obtained loadings were used to explain and to reconstruct mass balances' variance in the last century (since 1919). Results are statistically significant and pass the tests normally used in dendroclimatological reconstructions. They show variations of the mass balance compatible with known glaciological history in the Alps. Thus, authors conclude that the use of different wood-proxies permits the seasonal mass balance reconstruction of the Silvretta glacier.

The manuscript, in my opinion, is well written and the aims are clearly presented. Authors present exceptional datasets for an overlooked species in the Alps (i.e., Pinus cembra). In fact, in my knowledge, they present first isotope chronologies from Swiss stone pine in the area and one of the firsts chronologies of anatomical traits. Scientific design is solid and well presented. Moreover, only few dendroglaciological papers about European Alps were published, thus the manuscript is also characterized by a high level of novelty.

**REPLY:** We would like to very much acknowledge thereviewer for these words, the appreciation of the dataset and analyses as well as your detailed and critical assessment of our manuscript. We have taken all your suggestions into consideration, they helped to greatly improve the manuscript.

However, in my opinion, the use of some methodologies is partly questionable, and both discussion and conclusion lack a bit of control in some parts resulting presented in a bloated fashion and quite speculative way.

**REPLY:** We react to the specific comments below and try to address these general points raised here.

**SPECIFIC COMMENTS**

[#001] Line 25: δ13C isotope is not utilized for Bs reconstruction.

**REPLY:** We removed any reference to the δ13C isotope from the sentence which now reads as follows:
*"The combination of tree-ring width, radial cell wall thickness provide a highly significant reconstruction for summer mass balance, whereas, for winter mass balance, the correlation was less significant but still robust when radial cell lumen was combined with δ18O and δ13C records."*

[#002] Lines 35–36: Consider mentioning that glaciers are one of the Essential Climate Variables (ECV) as reported by the Global Climate Observing System (GCOS) (https://gcos.wmo.int/en/home)

**REPLY:** This reference to glacier as an ECV has been mentioned in the introduction section as follows:
*"Glaciers stand as one of the most important freshwater resources for societies and ecosystems. The recent increase in ice melt directly contributes to the rise of ocean levels. Recognizing their significance, the Global Climate Observing System (GCOS, https://gcos.wmo.int/en/home) has designated glaciers as an Essential Climate Variable (ECV). To reduce uncertainties in the quantification of future mass losses and their potential consequences, information on past glacier changes is essential as it allows improving simulations of past and future glacier evolution (e.g. Brunner et al., 2019)".*

[#003] Lines 57–66: Authors at lines 57–61 state that 'Mass balance modelling based on meteorological series (Huss et al., 2008; Nemec et al., 2009) offers an alternative method to infer glacier mass balance over long time-scales at high temporal resolution but results are not backed-up with in-situ observations before the onset of glaciological measurements and therefore might be biased or may incompletely resolve the relevant processes.'. At lines 62–66 they state: 'Tree-ring proxies clearly have the potential to overcome these limitations and to extend glacier mass balance series farther back in time. Based on the concept of Oerlemans and Reichert (2000) according to which mass balance series can be reconstructed from long meteorological records, several dendrochronological studies have been developed to demonstrate the reliability of high-elevation tree-ring proxies as reliable recorders of past summer temperature and – to a lesser extent – also winter precipitation (e.g., Büntgen et al., 2005; Coulthard et al., 2021; Carrer et al., 2023; Lopez-Saez et al., 2023).

These sentences seem in conflict to me. In the former authors declare that since there are not glaciological measures, reconstruction based on glaciological modelling and meteorological series might be biased, on the other hand, they declare, in the latter sentence, that tree-ring proxies can overcome to this limitation since they are representative of the summer temperature and winter precipitation that can be used to reconstruct mass balance. Thus, tree-ring proxies were used to reconstruct temperature and precipitation (i.e., meteorological series) that were used to reconstruct the mass balance that, however, supply results that might be biased since glaciological measurements are still missing before the start of the monitoring. Please consider clarifying the concepts.

REPLY: We agree that the ideas developed in this section were not clear to readers. We rephrased the paragraph entirely; it now reads as follows:
*"Mass balance modelling based on meteorological series (Huss et al., 2008; Nemec et al., 2009) allows inferring glacier mass balance over long time-scales at high temporal resolution. However, accurate modeling requires long records of temperature and precipitation from high-elevation meteorological stations located in the vicinity of glaciers, but such datasets are scarce. To address this limitation, meteorological series are generally scaled to the glacier sites (Huss et al., 2021). While air temperature show strong correlation over large distances (Begert et al., 2005) and, hence, allow for confident extrapolation, the distribution of precipitation in alpine environments is more difficult to estimate and larger uncertainties persist in winter mass balance reconstructions (Sold et al., 2016). High-elevation tree-ring proxies portray strong summer temperature and – to a lesser extent – winter precipitation signals over multi-centennial time periods (e.g., Büntgen et al., 2005; Coulthard et al., 2021; Carrer et al., 2023; Lopez-Saez et al., 2023). These are recognized as the main drivers of glacier fluctuations. Tree-ring proxies, located at high-elevation sites, in the vicinity of the glaciers, thus theoretically hold the potential to extend glacier mass balance series substantially farther back in time and thus offer an interesting alternative to meteorological series for mass balance reconstruction."*

[#004] Lines 77–78: In Cerrato et al. 2020 the authors did not used HISTALP dataset, but an improved version of the database presented in Brunetti et al. 2006 as described in material and methods. Brunetti, M., Maugeri, M., Monti, F., and Nanni, T.: Temperature and precipitation variability in Italy in the last two centuries from homogenised instrumental time series, Int. J. Climatol., 26, 345–381, https://doi.org/10.1002/joc.1251, 2006.

REPLY: That is a relevant feedback and we apologize if we created some confusion. The sentence was modified as follows:
*"More recently, Cerrato et al., (2020) reconstructed fluctuations of the Careser glacier (Italian Alps) since 1811 using MXD series and the long meteorological series available for the Alpine region (Brunetti et al., 2006, 2012, 2014; Crespi et al., 2018) to reconstruct summer and winter mass balance, respectively".*

[#005] Lines 91–93: In the manuscript only one glacier was tested and no comparison with others mass balance was performed. Thus, referring to 'Alpine glaciers' seems a bit pretentious if it is meant as glaciers of the entire European Alps.

REPLY: We agree and therefore nuance the sentence in the revision as follows:
*"This provides new insights into mass balance dynamics of an Alpine glacier during the maximum and termination of the Little Ice Age, a phase of important dynamics in glacier evolution but very limited direct evidence on the rates and the exact timing of changes".*

[#006] Line 98: Please consider adding the reference period for which the mean equilibrium line altitude is calculated. If it refers to the entire period (1919–2023) please consider adding also the maximum and the minimum since in the last century alpine glaciers withdraw quite continuously and abundantly.

REPLY: Thanks for the suggestion. Some precisions are added. We opt to only state the ELA over the period 1960-1990 as these years were characterized by relatively balanced mass budget:
*"The mean ELA of Silvretta was 2775 m a.s.l. between 1960 and 1990 with a standard deviation of 140 m."*

[#007] Line 108: the tree stand is located c. 30 km southeast of the glacier (not southwest).

REPLY: Thank you, the sentence was modified accordingly.

[#008] Line 149–150: Please check the total number of analysed rings. Between 1968 and 2017 there are 49 years, whereas between 1802 and 1967 there are 165 years, that divided by 5 results in 33 years (34 if 1802 is considered too). Thus, if I correctly understood, the total number of analysed rings is 83 and not 73 as reported.

REPLY: Thank you for the comment, the sentence was modified as follows:
*"The wood from each ring was processed separately between 1968 and 2017 and every fifth year between 1802 and 1967, so that in total, 873 years were measured on each individual core. For all other years between 1802 and 1965, material from the 10 trees of the same year was pooled prior to analysis."*

[#009] Line 164: Rsamp, samp with lowercase 's' to be coherent with the formula.

REPLY: The sentence was modified according to the comment

[#010] Lines 229–230: Please consider moving the sentence about the isotope at the end of the paragraph (at Line 243) to not interrupt the discussion about the QWA.

REPLY: The sentence was moved according to the comment. The paragraph now reads as follows:
*"This weaker signal is generally attributed to inter-annual variability in microscopic wood features (Olano et al., 2012; Liang et al., 2013; Pritzkow et al., 2014), heterogeneous intra-annual internal physiological processes which regulate carbon assimilation and allocation in tree rings (Eilmann et al., 2006; Fonti and García-González, 2004; Balanzategui et al., 2021) or to relationships with intra-annual environmental variables (Yasue et al., 2000; Ziaco et al., 2016) rather than to limiting factors exerted over the entire growing season (Eckstein, 2004; Ziaco et al., 2016). Regarding isotope parameters, Rbar values computed for 10-yr windows are 0.46 ($\delta 13C$) and 0.48 ($\delta 18O$)."*

[#011] Line 248: the number of the chapter is missing in section heading (i.e., 3).

*REPLY: Done*

[#012] Lines 260–261: probable typo, from July 15 to September 11 there are 58 days, not 152 as declared.

REPLY: There was clearly a typo in our sentence. There are indeed 58 days between July 15 to September 11.

Moreover, Figure 3A shows the mass balance reconstruction, I think that the figure was moved to supplementary material or totally removed.

REPLY: The sentence refers to Fig S1A and not to Fig 3A. It was therefore modified as follows:
*"In our case, highest correlations were obtained between the $CWT_{rad}$ chronology and temperatures over a 58-day time window extending from July 15 to September 11 (r=0.68; p<0.05, Figure S1A)."*

[#013] Line 267: see comment [#034].

REPLY: See answer to comment [#034].

[#014] Lines 269–270: The sentence is misleading. From Figure S1B very low values of correlation is interpretable, spanning between –0.1 and 0.1. In some situation, as function of the windows width, slightly positive or negative results can be appreciated (usually between (–)0.1 and (–)0.3).

REPLY: Fig S1B was very complicated for the reader to understand. We decided to remove it from the new version of the manuscript. Instead, it was replaced by a new figure (Fig. 3) which shows the correlations of each tree-ring parameter (TRW, Drad, CWTrad, $\delta$13C and $\delta$18O), daily temperature and precipitation for different subperiods (i.e. the growing season, ablation and accumulation periods).

[#015] Line 277: The winter signal embedded in $\delta$13C chronology is not reported in Table 2 and only partially inferable from supplementary material.

REPLY: Thank you for the comment. The fall (n-1)-winter signal embedded in $\delta$13C and $\delta$18O chronologies is now clearly highlighted in the new Fig. 3 and text was added to the manuscript. It has been precisely assessed in the new version of the manuscript as follows:
*"Both chronologies also portray a significant association with winter precipitation (October (n-1)-April (n), positive for $\delta$13C (r=0.19, p<0.01), and negative for $\delta$18O (r=-0.21, p<0.01)".*

[#016] Line 283–284: In table 3 only one combination for reconstructing Bs and Bw is reported. Consider rephrasing.

REPLY: The original sentence was rephrased as follows:
*"Based on this preliminary climate–growth relation analysis, we tested several combinations of parameters to reconstruct winter (Bw) and summer (Bs) glacier mass balances. Statistics obtained for the best combination of tree-ring proxies are reported in Table 3".*
The caption of Table 3 was also modified and now reads as:
*"Table 3. Statistics of Summer Mass Balance (SMB) and Winter Mass Balance (WMB) reconstructions based on the best combination of tree-ring proxies: TRW, CWTrad (for SMB) and $\delta$18O, $\delta$13C, Drad (for WMB) and their significance levels (p) at * p<0.05."*

[#017] Table 3: I wonder if all isotopes are really necessary in the Principal Component Regression model for Bw. More precisely, one isotope chronology (Carbon) resulted to be most sensible to the summer precipitation (i.e., ablation period, Table 2), and the other (Oxygen) is sensible to the precipitation from November to August (covering not only accumulation period but also quite the first two-third of the ablation season).

REPLY: We agree with the reviewer's comment. The Carbon isotope chronology is especially sensitive to summer precipitation. In addition, the inclusion of Carbon isotopic ratios in the principal component regression only affects the calibration/validation statistics of the winter mass balance reconstruction marginally (see new Table S1, added as supplementary material). For this reason, and following the reviewer's comment, we decided to disregard the $\delta$13C chronology for the winter mass balance reconstruction. For a more detailed response see comment [#20].

How different would be the results if only Drad is used? Or, in another way, are the authors sure to include proxies that are sensible to environmental parameters of ablation season in the reconstruction of Bw?

REPLY: Detailed statistics on mass balance reconstructions for each individual tree-ring proxy, as well as comprehensive information on all possible combinations of proxies, can be found in the newly added Table 3. The statistics confirm that the best combination of proxies for the summer mass balance reconstruction includes both TRW, $D_{rad}$ and $CWT_{rad}$, which are positively correlated with temperature of the ablation period (see new Fig. 3). In terms of statistics, the R2, RE and CE values increase from 0.38 to 0.47, 0.37 to 0.43 and 0.35 to 0.4, respectively, when both proxies are considered compared to using Drad alone.

[#018] Lines 289–293: Even if it is true that the first two Principal Components (PCs) positively correlate with Bw, seems that authors overlooked at the relationships between the original data and the PCs variables. If I correctly understood figure 3, it reported the correlation circle between the original variables (i.e., isotopes and Drad) and the first two dimensions obtained by PCA. Looking at that plot seems that only Drad is positively correlated with the PC's first dimension and basically uncorrelated with the second. I can thus hypothesize that, due to the

positive correlation index between the PC variables and the Bw (shown in Table 3), Drad is representative of the environmental condition that drive the Bw.

**REPLY:** We fully agree with the reviewer's comment. To clarify the relation between each tree-ring proxy, $B_w$ and $B_s$, we added a new Figure 3. Panel C clearly confirms the positive and significant correlation between $B_w$ and $D_{rad}$ (r=0.33, p<0.001) and that this proxy is representative of the environmental conditions that drive the $B_w$. Ecophysiologically, this correlation is thought to result from the link between water availability at the beginning of the growing season, which mostly depends on snowmelt, and cell enlargement.

Contrarily, Carbon isotope show a high negative correlation with PC first dimension (indicating a quite linear negative correlation with this dimension) with a (maybe significant) positive correlation with the second PC dimension. This seems to be coherent with previous analysis, i.e., Carbon isotope find its best correlation window with summer precipitation, and being summer precipitation mostly liquid, they enhance snow melting with heat transfer along the snowpack. Thus, probably, the negative correlation that was showed along the first PC dimension is representative not of a Bw, but of a Bs that, judging from figure 3, seems quite well negatively correlated with Bw.

**REPLY:** Figure 3B shows that $B_w$ is positively and significantly correlated to precipitation during the accumulation period (r=0.17, p<0.01 for the time window between Oct 1 and Apr 30 and 0.19 for the optimal period Oct 1-Feb 13). By contrast, the correlation between $B_w$ and $\delta^{13}C$ is negative and not significant (r=-0.09). These opposite correlations confirm, as underlined by the reviewer, that the $\delta^{13}C$ series only portrays a poor winter signal. Consequently, we removed this series from the revised $B_w$ multiproxy reconstruction and thank the referee for this very relevant and important feedback.

Considering the Oxygen isotopes, the series result completely uncorrelated with the PC first dimension being aligned with the axis of the second dimension. Considering that first and second dimension have quite the same explanatory power of the original dataset (i.e., 46.1 and 34.0, respectively) it is plausible that Oxygen represents something different from both Drad (precipitation from November to May and spring-summer temperature), and Carbon (summer precipitation and whole year temperature, this last point is questionable as reported in comments [#015] and [#017]). In fact, from previous analysis it results that Oxygen series is correlated with spring-summer temperature (as well as Drad) but with winter-to-summer precipitation. Maybe the representativeness of this variable of such a long precipitation period appoints it as a second major source of variability in the original dataset. Authors should consider these results, or at least supply more explanation on the motivation that drove them to use proxies sensible to summer precipitation and/or to two-third of the ablation season to reconstruct the Bw bearing in mind that correlation does not mean causation.

**REPLY:** The correlation between the $\delta^{18}O$ chronology, $B_w$ (r=-0.21, p<0.05) and winter precipitation (r=-0.21, p<0.01 for the time window between Oct 1 and Apr 30 and -0.22 for the optimal period Nov 29-Apr 2) is more consistent and more significant than the ones computed for $\delta^{13}C$.

Interestingly, many studies in the Tibetan Plateau and northwestern China (Grießinger et al. 2017; Wernicke et al. 2017; Liu et al. 2013; Qin et al. 2015; Xu et al. 2020), Northern Iran (Foroozan et al. 2020), Southern Kazakhstan (Qin et al., 2022) or Northern Pakistan (Treydte et al., 2006) reported that the oxygen isotope fractionation of tree ring is limited by winter precipitation. In Switzerland, at lower elevations (250 to 1850 m asl), Allen et al. (2019) demonstrated that trees frequently use winter-sourced water provided by snowmelt during the growing season and that tree-ring $\delta^{18}O$ values may thus reflect winter precipitation $\delta^{18}O$. In the Russian Arctic, Holzkamper and Kuhry (2009) suggested that the thickness of the snow and the timing of snow melt have a strong impact on the $\delta^{18}O$ composition of tree-ring α-cellulose because moisture in the early summer is most critical for wood formation. Soil and atmospheric drought caused by a deficit in previous winter alpine snowfall therefore lead to $\delta^{18}O$ enrichment in tree-ring α-cellulose.

Given this ecophysiological relationship between $\delta^{18}O$ composition and snow we decided to keep oxygen series as proxy for $B_w$ reconstruction. Table 2 shows that the inclusion of this proxy as predictor increase the robustness of the reconstruction (as compared to $D_{rad}$ only).

References:

Allen ST, Kirchner JW, Braun S, Siegwolf RTW, Goldsmith GR (2019) Seasonal origins of soil water used by trees. Hydrol Earth Syst Sci 23(2):1199–1210

Foroozan Z, Grießinger J, Pourtahmasi K, Bräuning A (2020) 501 years of spring precipitation history for the semi-arid Northern Iran derived from tree-ring δ18O data. Atmosphere 11:889

Griessinger J, Br.uning A, Helle G, Hochreuther P, Schleser G (2017) Late Holocene relative humidity history on the southeastern Tibetan plateau inferred from a tree-ring δ18O record: recent decrease and conditions during the last 1500 years. Quat Int 430:52–59

Holzkamper S, Kuhry P (2009) Stable isotopes in tree rings from the Russian Arctic—a proxy for winter precipitation? PAGES News 17(1):14–15

Liu H, Park Williams A, Allen CD, Guo D, Wu X, Anenkhonov OA, Badmaeva NK (2013a) Rapid warming accelerates tree growth decline in semi-arid forests of Inner Asia. Global Change Biol 19:2500–2510

Qin C, Yang B, Bräuning A, Grießinger J, Wernicke J (2015) Drought signals in tree-ring stable oxygen isotope series of Qilian juniper from the arid northeastern Tibetan Plateau. Glob Planet Chang 125:48–59

Qin L, Bolatov K, Yuan Y, Shang H, Yu S, Zhang T, Bagila M, Bolatov, A, Zhang R (2022) The spatially inhomogeneous influence of snow on the radial growth of Schrenk spruce (Picea schrenkiana Fisch. et Mey.) in the Ili-Balkhash Basin, Central Asia. Forests 13(1):44

Treydte KS, Schleser GH, Helle G, Frank DC, Winiger M, Haug GH, Esper J (2006) The twentieth century was the wettest period in northern Pakistan over the past millennium. Nature 440:1179–1182

Wernicke J, Hochreuther P, Grießinger J, Zhu H, Wang L, Bräuning A (2017) Multi-century humidity reconstructions from the southeastern Tibetan Plateau inferred from tree-ring δ18O Glob Planet Chang 149:26–35

Xu G, Liu X, Sun W, Szejner P, Zeng X, Yoshimura K, Trouet V (2020) Seasonal divergence between soil water availability and atmospheric moisture recorded in intra-annual tree-ring δ18O extremes. Environ Res Lett 15:094036

[#019] Lines 301–302: Authors hypothesis could be true, however should be noted that in the referred period occurred the last phase of positive Ba in the (Southern) Alps (Huss et al. 2015). Moreover, if the gap in Bw starts in 1984, I wonder how it is possible that the correlation values start to decrease 10 years before (as author stated, and as figure 4 shows, considering that the results are right-aligned, thus the considered 30-year window in 1974 is 1945–1974) and reach their lowest value nearby the years when the modelled Bw starts. Moreover, the lowering in correlation values ended around 2000 (i.e., the 30-year right-aligned window 1971–2000) when around 57% of the data are modelled. Considering this, the hypothesis supplied by authors seems to be not really supported by reported data. Probably, the changes in environmental conditions that bring less negative or even positive Ba, is not well represented by the selected variables for Bw (it is just a hypothesis that should be verified). On the other hand, a decrease of the correlation values in those years is observable also considering the seasonal mass balance reconstruction based on *Imfeld23* (both Bs and Bw) and the wood-proxy based Bs. Also in these cases, the lowering in correlation values starts well before the 1984, thus, in my opinion the lack of measured glaciological data could not be the (only) explanation to the observed behaviour in correlation trend. Huss, M., Dhulst, L., and Bauder, A.: New long-term mass-balance series for the Swiss Alps, J. Glaciol., 61, 551–562, https://doi.org/10.3189/2015jog15j015, 2015.

**REPLY:** We agree with the concern raised by the referee that the lack of measured glaciological data could not be the (only) explanation to the observed behavior in the correlation trend and that the shifts in environmental conditions leading to less negative or even positive winter balances may not be adequately captured by the chosen tree-ring proxies for the winter mass balance (Bw). To take this point into consideration, we have adjusted the original paragraph as follows:

*"Figure 5 shows the 30-yr moving correlations computed between the reconstructed and observed mass balances. Specifically, for winter, it shows a decrease of r values for time windows ending between 1981 and 2000 with r <0.25 for time windows ending between 1983 and 1999 (r<0.25, Figure 5A). Notably, this period coincides with the latter phase of positive Ba in the (Southern) Alps (Huss et al. 2015). One can therefore hypothesize that the shifts in environmental conditions that contribute to less negative or even positive annual glacier mass balances may not be adequately captured by the tree-ring proxies selected for winter balance reconstruction. Moreover, during a portion of this time period (1984-2003), no in situ measurements of winter mass balance were available (Huss and Bauder, 2009) and the gaps in the winter mass balance series were filled using a calibrated mass balance model driven by data from nearby meteorological stations (Huss et al., 2015). Therefore, it cannot be*

*ruled out that the decrease in correlation may be partly attributed to the quality of the mass balance time series rather than solely to the tree-proxy dataset.*
*For Bs, the 30-year correlations obtained from the observed and multi-proxy reconstructed summer mass balance time series consistently exceed 0.48 throughout the entire period and show limited standard deviation (0.06) between 1920 and 2017 (Figure 5A). However, while correlation values show an in increasing trend (from 0.52 to 0.74) for time windows ending before 2000, they significantly decrease reaching 0.50 by 2017. This reduction in prediction skill, from 0.74 to 0.50, starting in the 1970s, is comparatively less marked than the one documented by Cerrato et al. (2020) (0.45 to 0.2) for P. cembra, based on MXD records, albeit occurring a decade earlier".*

[#020] Lines 305–307: To me it is not clear the advantage in using all the variables deriving from a PCA instead of the original data. The PCA was thought to lowering the number of considered variables, creating new variables that 'summarize' the variance of the original data. Variables reduction is obtained retaining only those new variables that explain the largest part of the original data variance (usually 80% but it depends by the aims). If all PC variables are kept, it is equivalent to apply the multiparametric regression using the original data.

**REPLY:** The PCA was only used if the number of predictors included in the reconstruction exceeded 2. In other cases (i.e number of predictors ≤ 2), we only used multiple regression models. This was clearly stated in the methodology section as follows:
*"A multiple linear regression model was selected to reconstruct winter and summer mass balances. When more than two proxies were included in the model, the number of predictors were lowered using Principal Component Analysis (PCA), retaining the first n principal components (PCs) with eigenvalues exceeding 1".*

[#021] Line 308: consider modifying 'Ghiacciaio del Careser' to Careser glacier, as used before in the manuscript.

**REPLY:** Done

[#022] Lines 310–313: From figure 4 it is clear that the increasing trend appreciable from 1950 to 2000 (as exception of the 1980s where a decrease in correlation values is appreciable as commented in [#019]), is reverted to a negative trend since 2000 with correlation coefficient that drops from 0.75 to 0.5 in 17 years (mean decrease of -0.015 year-1, analysis should be performed to verify if the trends are significant and if the change is significant too, but I can speculate that, at least the negative one, is significant). Considering this, the decreasing of correlation starts in the 1970s, 10 years before the start of decreasing reported in Cerrato et al. 2020 (in Figure 4 the correlation are right-aligned, thus the 2000 value refers to 1971–2000 time window and this is why the decrease in correlation values starts in 1970s).

**REPLY:** We agree with the reviewer's comment and modified the section as follows:
*"For Bs, the 30-year correlations obtained from the observed and multi-proxy reconstructed summer mass balance time series consistently exceed 0.48 throughout the entire period and show limited standard deviation (0.06) between 1920 and 2017 (Figure 5A). However, while correlation values show an in increasing trend (from 0.52 to 0.74) for time windows ending before 2000, they significantly decrease reaching 0.50 by 2017. This reduction in prediction skill, from 0.74 to 0.50, starting in the 1970s, is comparatively less marked than the one documented by Cerrato et al. (2020) (0.45 to 0.2) for P. cembra, based on MXD records, albeit occurring a decade earlier."*

[#023] Lines 326–327: Being the correlation obtained using a multiple regression, this statement is not supported by data in this context, even if it is true as reported in cited papers. Moreover, consider to cite also Cerrato et al. 2019, that report data about the divergence between Swiss stone pine MXD and temperature in the high-frequency domain and being the source of data for Cerrato et al. 2020. Cerrato, R., Salvatore, M. C., Gunnarson, B. E., Linderholm, H. W., Carturan, L., Brunetti, M., De Blasi, F., and Baroni, C.: A Pinus cembra L. tree-ring record for late spring to late summer temperature in the Rhaetian Alps, Italy, Dendrochronologia, 53, 22–31, https://doi.org/10.1016/j.dendro.2018.10.010, 2019.

**REPLY:** We agree with the reviewer's comment and modified the initial sentence as follows:
*"These results confirm that our multiproxy reconstruction records only suffer from very limited divergence and standardization issues which notoriously affect both TRW and MXD records (Cook et al., 1995; Björklund et al., 2019; Cerrato et al., 2019)."*

[#024] Lines 351–353: A reconstruction of an Alpine glacier mass balance at annual scale was already reported by Cerrato et al. 2020 and by Nicolussi and Patzelt, 1996 (in my knowledge, but could be other studies. These studies are already cited in the manuscript even if the latter is missing in the reference list) and both show less negative (or even positive) mass balances around the last peak of the LIA. Please consider rephrasing. Moreover, in the present study, volcanic forcing or radiative data were not considered, thus the sentence, in the present form, seems a bit speculative to me.

REPLY: We agree with the referee that the sentence was too speculative as we did not analyze in detail (i.e. at the annual scale and for each volcanic eruption) the impact of volcanic forcing on glacier mass balance. We therefore removed the sentence in the revised version of the manuscript.

[#025] Lines 354–364: Speculative paragraphs. If two reconstructions are available and no verification is possible, it is basically impossible to determinate which is the most correct. It is certainly true that back in time, meteorological series loss representativeness and explanatory power in remote areas, but, considering reported data, also tree-ring proxies reconstruction suffer of a decrease of explanatory power in cold phases (see for instance 1980s where mass balance data, even if modelized, are present) and also after 2000s (even if in this last case the environmental conditions that drive a loss of correlation are hotter than the previously experienced; see comments [#019] and [#022] for more details). Thus, concluding that the Ba based on *Imfeld23* lacks representativeness whereas tree-ring bases reconstruction surely represent the behaviour of the glacier in such bloated form seems a bit speculative. Moreover, authors never consider that their approach using the meteorological data could suffer of a big and simply issue: authors calibrated the reconstruction using an optimal time window based on temperature and precipitation occurred since 1919. Statistics are solid and tests were passed in the considered period. By counterpart, authors are assuming, based on their results, that the length of the accumulation and ablation seasons are the same in a period where the temperature has been proved been lower and also precipitation might be, testified by a different duration of the snowpack (Carrer et al., 2023, cited in the manuscript). Maybe authors should consider that is not the meteorological dataset, but the selected optimal window of a fixed length based on recent environmental conditions that can bias the results, as already reported in Cerrato et al. 2020. However, should be noted that the here proposed reconstruction for Bw shows lower values during the Dalton minimum compatible with previous work that reported more dryer winter during that period (Anet et al., 2014). However, the *Imfeld23* based Bw is not reported, thus it is impossible to evaluate if also meteorological-based Bw reconstruction shows comparable results. Anet, J. G., Muthers, S., Rozanov, E. V., Raible, C. C., Stenke, A., Shapiro, A. I., Brönnimann, S., Arfeuille, F., Brugnara, Y., Beer, J., Steinhilber, F., Schmutz, W., and Peter, T.: Impact of solar versus volcanic activity variations on tropospheric temperatures and precipitation during the Dalton Minimum, Clim. Past, 10, 921–938, https://doi.org/10.5194/cp-10-921-2014, 2014.

REPLY: We would like to thank the reviewer for this very valuable comment. We agree with the hypotheses that were formulated to explain the difference between the multiproxy and *Imfeld23* reconstructions since 1850. Following the reviewer's recommendation, we modified this section as follows:
*"One could speculate that this divergence between the two reconstructions during preindustrial times could be attributed (i) to tree-ring proxy, particularly their reduced explanatory power in colder periods as evidenced for time windows ending between 1983 and 1999 (see §3.3). The divergence could also stem (ii) from the optimal fixed-length window utilized in the Imfeld23 reconstruction, calibrated on recent environmental conditions. This methodology assumes a constant length for the accumulation and ablation seasons since the early 19th century, despite significant variations in temperature, precipitation, and snowpack conditions compared to the present (Carrer et al., 2023). Such an assumption may introduce bias into the reconstruction (Cerrato et al., 2020). Finally, (iii) the complete absence of high-elevation records available in the Imfeld23 dataset prior to 1864 (see methods) raises questions about the robustness of the reconstruction. We cannot exclude that the gridded temperature and precipitation fields might fail to accurately reproduce changes in winter precipitation distributions in the early stages of the reconstruction."*

[#026] Lines 372–374: The study can be considered a first step in Switzerland, but not throughout the Alps since both Cerrato et al. 2020 and Nicolussi and Patzelt, 1996 already presented mass balance reconstructions. Consider rephrasing.

REPLY: The sentence was rephrased as follows:

*"The study also constitutes an important step in extending glacier mass-balance records beyond the instrumental period for the Swiss Alps".*

[#027] Lines 374–375: Due to the lacks of validation on the correctness of the Ba reconstructions (at the actual state it is impossible to define which is the most correct reconstruction between the wood-proxy based and the *Imfeld23*-based Ba since no comparison with previous reconstruction is performed, neither a comparison between the potential glacier volume with geomorphological and/or cartographical evidence) the sentence seems quite speculative.

**REPLY:** We agree that this sentence could be considered speculative and therefore removed it from the revised version.

[#028] Lines 378–380: This sentence seems speculative. Since it is impossible to validate both wood-based and *Imfeld23* reconstructions (see comments [#025] and [#027]) it is also impossible to be sure of the correctness of the estimated quantity of water equivalent gain (or loss) in period were the reconstructions differ in a more pronounced way (from Figure 4 and 5, the period of less agreement between wood-proxies based and *Imfeld23*-based reconstructions occurred for the entire XIX Century). Moreover, Authors stated earlier that in the earlier portion the used meteorological dataset is not completely reliable, so it is impossible to verify this sentence, maybe the disagreement between the expected and obtained *Imfeld23* Ba is due to the uncertainties of the original dataset, or maybe not.

**REPLY:** The original sentence was rephrased in a more nuanced way as follows*:*
*"Our results based on multiple wood-proxies reveal that glacier mass gains during the final stages of the Little Ice Age were strongest between 1810 and 1820. Considering the synchronicity of increasing mass balance with a cluster of volcanic eruptions and diminished solar activity, we align with Sigl et al. (2015, 2018) in hypothesizing that these gains may partly result from the co-occurrence of volcanic forcing and the Dalton Minimum".*

[#029] Figure 1: Consider inverting the vertical order of the inset A and both B and C.

**REPLY:** As suggested, the insets A and B-C have been inverted

[#030] Figure 2: Consider to explain the meaning of the purple dotted line (or purple dots, but it seems a line to me) in caption.

**REPLY:** The caption was modified as follows:
*"Figure 2. Profiles of (A) radial cell-wall thickness (CWTrad) and (B) radial cell diameters (Drad) along P. cembra tree rings. Purple dots represent the mean values of twenty trees over 217 years (1800–2017) smoothed using a polynomial regression (black line) represented with its 95th confidence interval (shadowed purple areas). The blue line represents maximum values for each of the wood parameters analyzed for 40 µm wide radial bands. The dotted black line shows the mean relative position of the transition between earlywood and latewood according to Morck's index = 1."*

[#031] Figure 3: caption, unclear to me, please consider rephrasing.

**REPLY:** The caption was modified as follows:
*"Figure 3. Winter (A) and summer (B) mass balance of the Silvrettagletscher reconstructed from tree-ring proxies over the 1802-2016 period. The thin light blue and orange curves illustrate interannual variations in winter and summer mass balance, respectively, derived from δ18O and Drad (winter) and TRW, CWTrad,and Drad (summer). The dark blue and dark brown curves represent Silvrettagletscher's winter and summer mass balance records from 1919-2016. Thick lines indicate decadal variations, smoothed using an 11-year spline".*

[#032] Figure 5: please consider to maintain the same y-axis scale among the plots for readability.

**REPLY:** The same y-axis limits (from -3 to -1) have been used for the three panels of the new figure (now Fig. 6).

[#033] Figure S1: caption is misleading on the time-window information. Moreover, reported information (e.g., standardization method and windows length) does not match those declared in the main manuscript and thus

the results are not easily comparable with those reported in the main text. In fine, the addition of contour lines at the significance level of p<0.05 will be appreciated.

**REPLY: F**igure S1 has been removed from the manuscript and all the information initially contained in this figure are now synthesized in Fig. 3

Figure S1. (A) Correlations of the raw radial cell-wall thickness ($CWT_{rad}$) and detrended tree-ring width (TRW) chronologies with air temperatures reconstructed with the Imfeld et al. (2023) dataset over time windows ranging from 121-to 273-day windows. (B) Correlations of the raw $\delta^{18}O$, $\delta^{13}C$ and $D_{rad}$ chronologies and *Imfeld23* precipitation sums over time windows ranging from 274 (n-1) to 120-day windows.

**REPLY: F**igure S1 has been removed from the manuscript and all the information initially contained in this figure are now synthesized in Fig. 3

[#034] Table 2: correlation between δ13C and temperature: the reported optimal time window is equal in length to the maximum window tested (330 days), probably enlarging the tested windows, other (and longer) 'optimum windows' could be found. Beside this mine speculative consideration, authors in M&M stated that they '*calibrated regression models on temperature and precipitation averaged over 30 to 330-day windows starting on October 1 of the year preceding ring formation (n-1) and ending on September 30 of the year in which the ring was formed (n)*'. Results does not match with declared methods. Caption: 'optimal time windows used in annual mass balance reconstructions' is misleading. If I correctly understood, these are the optimal time windows resulting from the correlation analysis between the tree-ring parameters and the meteorological series. Mass balances are not involved in these results.

**REPLY:** We agree with the reviewer that the caption of Fig. 2 was erroneous. The initial Table 2 has been profoundly reworked. Correlation analyses are detailed in Fig. 3 which synthesized in panels A and B the correlations between the tree-ring parameters and climate variables (temperature and precipitation) for the growing season, ablation and accumulation periods. In this figure, the optimal time windows are highlighted precisely. The correlations between tree-ring parameters, winter and summer glacier mass balance series are given in Fig. 3B.

[#035] Table 3: consider using Bs, Bw, CWTrad, Drad, δ13C, and δ18O both in table and in caption to be coherent with the rest of the manuscript.

**REPLY:** The table has been modified according to the reviewer's suggestion

**Response to Reviewer #2.**

Dear Editor,

Below we provide a point-by-point response to the comments of Reviewer 2 that were very helpful to finalize the manuscript. Our responses to the reviewer comments are given in *orange*, and new text in the manuscript is pasted in quotation marks.

The authors use multiple proxies from Pinus cembra trees from God da Tamangur to reconstruct seasonal glacier mass balance for the nearby Silvrettagletscher over the last two centuries. They combine tree-ring width, radial cell wall thickness, and δ13C isotope records to reconstruct summer mass balance, and radial cell lumen, δ18O, and δ13C records to reconstruct winter mass balance.

The manuscript presents an interesting and new tree-ring time series which is valuable information in the research domain, especially since only a few papers regarding the reconstruction of glacier mass balance based on tree-ring parameters were previously published. The paper is novel and has original elements, still, some of the discussions are speculative, and major revisions are required before further decisions.

*Reply: We would like to thank the reviewer for the careful reading of our manuscript. We have taken all your suggestions into consideration. They were very helpful to finalize the manuscript.*

**General and specific comments:**

The title of the paper is very long, please consider adjusting it.
Even if in the title authors indicate *"Abrupt termination of the Little Ice Age",* this aspect is very shortly discussed in the paper, and only one paragraph is dedicated to this aspect.

**Reply:** *The title was shortened and modified to be in line with the aim of the paper. It now reads as follows:*
*"Multiproxy Tree-Ring Reconstruction of Glacier Mass Balance: Insights from Pinus cembra Trees near Silvrettagletscher in the Swiss Alps"*

The method section is very poorly organized and presented, and it is very hard to follow the presented data and methods, major improvements are required.

**Reply:** *We have made an effort to improve the methods part as much as possible, also relying on the feedback provided by referee #1 and the feedback provided by reviewer #2 in the following.*

Also, in the method section, add in a succinct but clear way, information about how many samples were used for every tree-ring parameter, and what is the measured time span for every tree-ring parameter.

**Reply:** *We now state for TRW that:*
*"Tree cores were collected during a field campaign in summer 2018. To perform tree-ring width (TRW), 46 trees were sampled using a 12 mm increment borer. From each tree, we extracted two increment cores at breast height (c. 130 cm above ground). Ring widths were measured to the nearest 0.01 mm using TSAPWin (Rinntech, Germany), cross-dated using standard dendrochronological procedures (Stokes and Smiley, 1996) and checked for dating and measurement errors with the COFECHA software (Holmes, 1983). Ring widths from single radii were summarized to mean widths per tree. Values from 20 individual trees showing the best TRW inter-series correlation and covering the period 1802–2017, in order to ensure consistent sample depth across time, were averaged into a master TRW chronology.".*
*For QWA:*
*"To perform wood anatomical analyses, the first of the two sampled cores from each of the 20 individuals included in the master TRW chronology was split into 4–5 cm long pieces to obtain 15 μm thick cross-sections with a rotary microtome (Leica RM 2255/2245). The sections were stained with Safranin and Astra blue to increase contrast and fixed with Canada balsam following standard protocols (Gärtner and Schweingruber, 2013; von Arx et al., 2016). Digital images of the microsections - at a resolution of 2.27 pixels/μm - were produced at the Swiss Federal Research Institute WSL (Birmensdorf, Switzerland), using a Zeiss AxioScan Z1 (Carl Zeiss AG, Germany). For the 20 trees, we used the ROXAS (v3.1) image analysis software (von Arx and Carrer, 2014) to automatically detect anatomical structures for all tracheid cells and annual ring boundaries for the period 1800–2017".*

*For isotopic analyses:*
*For the isotopic analyses ($\delta^{18}O$ and $\delta^{13}C$), we selected ten trees showing the best inter-series correlation out of the 20 trees used for TRW and QWA analyses. Selected samples aged between 242 and 634 years old at the time of sampling"*

A figure with measured raw tree-ring data (all parameters) and their replication is necessary.

**Reply:** Figure S1 has been added in order to present the raw data. The replication is not provided in the figure as it does not change across time as mentioned in the material and methods section:
*"Values from 20 individual trees showing the best TRW inter-series correlation and covering the period 1802–2017, in order to ensure consistent sample depth across time, were averaged into a master TRW chronology."*

The aim of the paper needs revisions. The present aim does not reflect the present title of the manuscript and it is not clear what are the objectives of this manuscript.

**Reply**: The aim of the paper was changed as follows:
*"The aim of this study is to assess the reliability of a multiproxy approach, using only tree-ring proxies, in extending historical seasonal mass balance data, including winter and summer mass balance series, into the past. To reach this goal, we employed stable isotope ($\delta18O$, $\delta13C$) and tree-ring anatomy chronologies of P. cembra which has recently been shown to be very sensitive to mean temperature over the ablation season (April–September; Lopez-Saez et al., 2023). We selected Silvrettagletscher in the Eastern Swiss Alps as our study site due to the availability of glacier mass balance data spanning from 1920 to present, making it one of the longest continuous series in the Alps.*

Please check all the abbreviations in the manuscript, including tables and figures captions.
**Reply** : All the abbreviations have been carefully checked.

Section 2.6 Climate–growth relationships, need major revision. The presented information here is hard to follow and it is not complete. It is not clear between which climate parameters and which tree-ring parameters were made correlation analyses.
**REPLY:** We fully agree with the reviewer's comment. To clarify the relation between each tree-ring proxy, $B_w$ and $B_s$, we added a new Fig. 3. Panels A and B the correlations between the tree-ring parameters and climate variables (temperature and precipitation) for the growing season, ablation and accumulation periods. In this figure, the optimal time windows are highlighted precisely. The correlations between tree-ring parameters, winter and summer glacier mass balance series are given in Fig. 3B. The Section 2.6 has been profoundly modified based on results provided in this new figure.
.
The link between mass balance and variability of the tree ring parameters is not presented, and this is a key point for the presented reconstruction and needs explanations.
**REPLY:** The correlation between mass balance and variability of the tree ring parameters is now presented in panels C and D of the newly added Fig. 3.

Line 219 " all samples" it is confusing, please rephase
 **Reply**: The sentence was rephrased as follows:
*"To perform wood anatomical analyses, the cores of the 20 individuals included in the master TRW chronology were split into 4–5 cm long pieces to obtain 15 µm thick cross-sections with a rotary microtome (Leica RM 2255/2245). The sections were stained with Safranin and Astra blue to increase contrast and fixed with Canada balsam following standard protocols (Gärtner and Schweingruber, 2013; von Arx et al., 2016). Digital images of the microsections - at a resolution of 2.27 pixels/µm - were produced at the Swiss Federal Research Institute WSL (Birmensdorf, Switzerland), using a Zeiss AxioScan Z1 (Carl Zeiss AG, Germany). For the 20 trees, we used the ROXAS (v3.1) image analysis software (von Arx and Carrer, 2014) to automatically detect anatomical structures for all tracheid cells and annual ring boundaries for the period 1800–2017. We excluded measurements of samples with cell walls damaged during sampling or preparation and focused on two parameters in quantitative wood anatomy analyses: radial cell lumen diameter ($D_{rad}$) and radial cell wall thickness ($CWT_{rad}$) (Prendin et al., 2017; von Arx and Carrer, 2014)."*
*"*

The correlation values based on the presented color bar are impossible to distinguish, and the figure needs major improvements.

**Reply:** In order to increase the readability, Figure S1 has been removed and all the correlation are now presented in the new Fig. 3

Line 267-269, please consider adding a graph to present the correlation coefficients presented here, and to improve this paragraph because it is hard to understand what the authors want to present here.

Reply: The new Fig. 3 has been added following the reviewer's suggestion. The paragraph has been fully reworded, considering the comment of Referee#1. It now reads as follows:

*"$\delta^{13}C$ chronologies are negatively correlated with mean daily temperature from October (n-1) to September (n) (r=-0.4, p<0.001) and especially from October 8 (n-1) to May 7 (n) (r=-0.42, p<0.001). Mean daily temperature from October (n-1) to September (n) (r=0.36, p<0.001) and during the growing season (April 11-September 14, r=0.44, p<0.001) are the main drivers of $\delta^{18}O$ variations. A negative correlation is also found between $\delta^{13}C$ and May 26–July 26 (n) (r=-0.22, p<0.01) and between $\delta^{18}O$ and fall (n-1) to summer (n) precipitation totals (r=-0.25, p<0.01) (Fig.3). Both chronologies also portray a significant association with winter precipitation October (n-1)- April (n), positive for $\delta^{13}C$ (r=0.17, p<0.01), and negative for $\delta^{18}O$ (r=-0.21, p<0.01)."*
*"*

Different colors must be used for Figures 3A and 3B, it is impossible to distinguish the difference between two blue colors or two orange colors.

**Reply:** The contrast between the two blue and orange colors has been increased in Fig. 3A, B.

Line 305-315 need to be rephrased and improved.

**REPLY:** We agree with the reviewer comment and modified the section as follows:

*"For Bs, the 30-year correlations obtained from the observed and multi-proxy reconstructed summer mass balance time series consistently exceed 0.48 throughout the entire period and show limited standard deviation (0.06) between 1920 and 2017 (Figure 5A). However, while correlation values show an in increasing trend (from 0.52 to 0.74) for time windows ending before 2000, they significantly decrease reaching 0.50 by 2017. This reduction in prediction skill, from 0.74 to 0.50, starting in the 1970s, is comparatively less marked than the one documented by Cerrato et al. (2020) (0.45 to 0.2) for P. cembra, based on MXD records, albeit occurring a decade earlier."*

The differences between proxy mass balance and imfeld23 mass balance are huge, and from the presented text it is not clear why and which time series should we trust. Section 3.4 is rather speculative. Please consider eliminating the speculative affirmations.

**REPLY:** Section 3.4 has been profoundly modified following the recommendation of both reviewers. The speculative affirmations have been removed (see answer to comments [25] and [28] of reviewer#1) . In addition, the difference between *Imfeld23* and our multiproxy reconstructions are discussed in more detail as follows:

*"One could speculate that this divergence between the two reconstructions during preindustrial times could be attributed (i) to tree-ring proxy, particularly their reduced explanatory power in colder periods as evidenced for time windows ending between 1983 and 1999 (see §3.3). The divergence could also stem (ii) from the optimal fixed-length window utilized in the Imfeld23 reconstruction, calibrated on recent environmental conditions. This methodology assumes a constant length for the accumulation and ablation seasons since the early 19th century, despite significant variations in temperature, precipitation, and snowpack conditions compared to the present (Carrer et al., 2023). Such an assumption may introduce bias into the reconstruction (Cerrato et al., 2020). Finally, (iii) the complete absence of high-elevation records available in the Imfeld23 dataset prior to 1864 (see methods) raises question about the robustness of the reconstruction. We cannot exclude that the gridded temperature and precipitation fields might fail to accurately reproduce changes in winter precipitation distributions in the early stages of the reconstruction."*

---

## Author Response (AR2)

**Response to Reviewer #1. Riccardo Cerrato**

Dear Editor,

Below we provide a point-by-point response to the comments of Reviewer 1 that were very helpful to finalize the manuscript. Our responses to the reviewer comments are given in *orange*, and new text in the manuscript is pasted in quotation marks.

**GENERAL COMMENT**

In the manuscript 'Abrupt termination of the Little Ice Age in the Alps in the mid-19th century: lessons from a multi-proxy tree-ring reconstruction of glacier mass balance' Lopez-Saez and co-authors present seasonal (and annual) mass balance reconstructions for a Swiss glacier since 1802 CE. Authors use several proxies obtained by different methods (total ring width, quantitative wood analysis, and isotopes) and Principal Component Analysis to perform a multiparameter linear regression. The obtained scores were used to explain and to reconstruct mass balances' variance in the last century (since 1919). Results are statistically significant and pass the tests normally used in dendroclimatological reconstructions. They show variations of the mass balance compatible with known glaciological history in the Alps. Thus, authors conclude that the use of different wood-proxies permits the seasonal mass balance reconstruction of the Silvretta glacier. The manuscript, in my opinion, is well written and the aims are clearly presented. Authors present exceptional datasets for an overlooked species in the Alps (i.e., Pinus cembra). In fact, in my knowledge, they present first isotope chronologies from Swiss stone pine in the area and one of the first chronologies of anatomical traits. Scientific design is solid and well presented. Moreover, only few dendroglaciological papers about European Alps were published, thus the manuscript is also characterized by a high level of novelty. I had the opportunity to read a previous version of the manuscript, and I can appreciate the amount of work performed by the authors to clarify some points. Their replies are convincing and pertinent; the manuscript was amended accordingly. Only a few typos are still present and they are reported below.

**REPLY:** We would like to very much acknowledge the reviewer for these words. We have taken all your suggestions into consideration, they helped to greatly improve the manuscript.

**SPECIFIC COMMENTS**

[#001] Page 1 line 23: maybe typo, I think that "s.le" means "stable"

**REPLY:** Thank you, the sentence was modified accordingly.

[#002] Page 1 line 23 (and following occurrences at line 330, 331, and 345): the abbreviation of Latin locution "id est" usually is followed by a comma.

**REPLY:** Thank you, sentences were modified accordingly.

[#003] Page 2 line 44: as well as "id est", also the abbreviation of "exempli gratia" is usually followed by a comma.

**REPLY:** Thank you, the sentence was modified accordingly.

[#004] Page 3 line 81: formatting typo (dashed underlined reference).

**REPLY:** Thank you, the reference was modified accordingly.

[#005] Page 3 line 84: in the reference list "Cerrato et al. 2020" is missing.

**REPLY:** Thank you, the reference was added accordingly.

[#006] Page 4, 12, 14, and 15 lines 106, 277, 327, 334, and 337: All along the manuscript the glacier was identified as "Silvrettagletscher" as exception of five times that was identified as "Silvretta glacier". Consider changing the latter for coherence.

**REPLY:** Thank you, we changed "Silvrettagletscher" by "Silvretta glacier" throughout the manuscript.

[#007] Page 7 line 179: Please consider moving the reference list at the end of the sentence.

**REPLY:** Thank you, we moved the reference list at the end of the sentence.

[#008] Page 7 line 187: Actually, R Studio is a GUI for the program language R. Consider to cite also the R-project: R Core Team, 2023. R: A Language and Environment for Statistical Computing. R Foundation for Statistical Computing, Vienna, Austria. https://www.R-project.org/., R Foundation for Statistical Computing, Vienna, Austria version: 4.3.1 (2023-06-16)-Beagle Scouts [*Refer to the used version of R*]). Moreover, the reference is missing in the reference list.

**REPLY:** Thank you, we changed the reference and added in the reference list.

[#009] Page 14 line 316: Authors report and discuss the results of the analysis performed on δ13C series in the previous paragraph, results that, in my opinion, are worthy to be reported. However, in this and in the following paragraphs the results are completely ignored with no explanations to the readers that did not read previous version of the manuscript or the comments in the on-line discussion. I think that a single sentence that explains the motivations of the exclusion of δ13C series from the following analysis is necessary.

**REPLY:** Thank you, we agree and we added a single sentence as follow:
.
"The isotopic parameter δ13C has been excluded from the combinations for Bw and Bs reconstruction because statistically, it is not significant".

[#010] Page 14 line 320: Typo in "Statistics or these reconstructions are reported in Tab. 3."

**REPLY:** Thank you, we agree and we change "or" by "of".

[#011] Page 15 line 344: I think that the given reference "Holzkämper and Kuhry, 2009" should be "Cerrato et al. 2020", instead.

**REPLY:** Thank you, we agree and we changed the renfence Holzkämper and Kuhry, 2009" by "Cerrato et al. 2020.

[#012] Page 18 line 411: "[…] decrease significantly before the 1860s (Fig. 5B)" maybe authors mean 1960s, instead. In Fig. 5B 1860s are out of the range, otherwise should be reported that with 1860s is intended the start of the moving window.

**REPLY:** Thank you, we agree and we change "1860" by "1960".

Figure 2: Caption (line 256): "The dotted black line […]" I think that in the amended figure the line has become solid and red.

**REPLY:** Thank you, we agree and change the caption.

Figure 3: Caption (line 277): see comment [#006]

*REPLY: Done*

Figure 4: Please consider specifying what the horizontal line represents (I suppose the mean of the reconstructed series).

**REPLY:** Thank you, we agree and we added the caption sentence as follow:
"The black line represents the mean of the Winter (A) and summer (B) mass balance reconstructed series."